# TMC1 is an essential component of a leak channel that modulates tonotopy and excitability of auditory hair cells in mice

**Shuang Liu[1,2†], Shufeng Wang[1,2†], Linzhi Zou[1,2†], Jie Li[1,2], Chenmeng Song[1,2], Jiaofeng Chen[1,2], Qun Hu[1,2], Lian Liu[1,2], Pingbo Huang[3,4,5], Wei Xiong[1,2*]**

[1]School of Life Sciences, Tsinghua University, Beijing, China; [2]IDG/McGovern Institute for Brain Research at Tsinghua University, Beijing, China; [3]Department of Chemical and Biological Engineering, Hong Kong University of Science and Technology, Hong Kong, China; [4]State Key Laboratory of Molecular Neuroscience, Hong Kong University of Science and Technology, Hong Kong, China; [5]Division of Life Science, Hong Kong University of Science and Technology, Hong Kong, China

**Abstract** Hearing sensation relies on the mechano-electrical transducer (MET) channel of cochlear hair cells, in which transmembrane channel-like 1 (TMC1) and transmembrane channel-like 2 (TMC2) have been proposed to be the pore-forming subunits in mammals. TMCs were also found to regulate biological processes other than MET in invertebrates, ranging from sensations to motor function. However, whether TMCs have a non-MET role remains elusive in mammals. Here, we report that in mouse hair cells, TMC1, but not TMC2, provides a background leak conductance, with properties distinct from those of the MET channels. By cysteine substitutions in TMC1, we characterized four amino acids that are required for the leak conductance. The leak conductance is graded in a frequency-dependent manner along the length of the cochlea and is indispensable for action potential firing. Taken together, our results show that TMC1 confers a background leak conductance in cochlear hair cells, which may be critical for the acquisition of sound-frequency and -intensity.

DOI: https://doi.org/10.7554/eLife.47441.001

*For correspondence:
wei_xiong@tsinghua.edu.cn

†These authors contributed equally to this work

## Introduction

Hair cells are mechanoreceptors that convert mechanical stimuli provided by sound and acceleration into electrical signals. In the snail-shaped mammalian cochlea, hair cells are organized into three rows of outer hair cells (OHCs) and one row of inner hair cells (IHCs) that run along the length of the cochlear duct. The cochlea is tonotopically organized, where hair cells at the base of the cochlea signal high-frequency sounds and hair cells at the apex catch low-frequency sounds, with a gradient in between. OHCs amplify input sound signals whereas IHCs transmit sound information to the central nervous system (CNS).

The mechanotransduction complex in cochlear hair cells consists of a multitude of proteins, including ion channel subunits, cell adhesion proteins, myosin motors, and scaffolding proteins that are critical to sense sound-induced force (*Xiong and Xu, 2018*). The transmembrane proteins TMC1, TMC2, lipoma HMGIC fusion partner-like 5 (LHFPL5), and transmembrane inner ear expressed protein (TMIE), are thought to be integral components of the MET channels in hair cells. TMC1 and TMC2 have been proposed to be the pore-forming subunits of the MET channel in hair cells (*Ballesteros et al., 2018*; *Corey and Holt, 2016*; *Kawashima et al., 2015*; *Pan et al., 2018*). Consistent with this model, MET currents are absent in hair cells from mice lacking both TMC1 and TMC2 (*Kawashima et al., 2011*), while the unitary conductance, permeability, and ion selectivity of the

MET channel differs between hair cells expressing only TMC1 or TMC2 (*Beurg et al., 2015a*; *Beurg et al., 2014*; *Corns et al., 2017*; *Corns et al., 2014*; *Corns et al., 2016*; *Kim and Fettiplace, 2013*; *Pan et al., 2013*). Finally, cysteine mutagenesis experiments are consistent with the model that TMC1 is a pore-forming subunit of the hair-cell MET channel (*Pan et al., 2018*). However, all efforts have so far failed to express TMC proteins in heterologous cells to reconstitute ion channel function (*Corey and Holt, 2016*; *Wu and Müller, 2016*). Interestingly, MET responses in OHCs vary tonotopically, and a lack of TMC1 and LHFPL5, but not TMC2, abolishes the tonotopic gradient in the MET response (*Beurg et al., 2014*; *Beurg et al., 2015b*). While changes in the levels of expression of TMC1 from the base to the apex have been proposed to underlie the tonotopic gradient in the MET response, the mechanisms that cause the tonotopic gradient are not completely defined (*Beurg et al., 2018*; *Beurg et al., 2006*; *Ricci et al., 2003*; *Waguespack et al., 2007*).

TMC orthologues in other species have been linked to a diversity of functions. In *Drosophila melanogaster*, TMC is expressed in the class I and class II dendritic arborization neurons and bipolar dendrite neurons, which are critical for larval locomotion (*Guo et al., 2016*). TMC is also enriched in md-L neurons that sense food texture (*Zhang et al., 2016*), and for proprioceptor-mediated direction selectivity (*He et al., 2019*). In *Caenorhabditis elegans*, TMC1 regulates development and sexual behavior (*Zhang et al., 2015*), and is required for the alkaline sensitivity of ASH nociceptive neurons (*Wang et al., 2016*). While efforts have failed to demonstrate that TMCs in flies and worms are mechanically gated ion channels, recent mechanistic studies in worms have shown that TMC1 and TMC2 regulate membrane excitability and egg-laying behavior by conferring a leak conductance (*Yue et al., 2018*). This raises the question of whether mammalian TMC1 and TMC2 only function as components of mechanically gated ion channels, or possess additional roles critical for mechanosensory hair-cell function.

In this study, we set out to determine the non-MET functions of TMCs and to tackle the link with hair-cell function by manipulating TMCs genetically and monitoring membrane current and potential in mouse hair cells. We sought out potential molecular and cellular mechanisms underlying TMCs and their correlated relevance in auditory transduction.

## Results

### TMC1 but not TMC2 mediates a background current in hair cells

During the whole-cell voltage-clamp recording from P6 outer hair cells (OHCs) (*Figure 1A*) in regular 144 mM $Na^+$-containing external solution (144 Na), we always recorded a 'leaky' membrane current ($I_m$, 73 pA on average) (*Figure 1B,C*). When $Na^+$ was replaced in the external solution by N-methyl-D-glucamine ($NMDG^+$) (144 NMDG), the $I_m$ was small (*Figure 1B*), demonstrating that this background current is significantly carried by an ion channel in the cell membrane. When reperfused with 144 Na solution, the current baseline returned to 'leaky' status (*Figure 1B*). However, the $I_m$ was markedly diminished in *Tmc1*-knockout OHCs (*Figure 1B,C*). For more accurate quantification, the amplitude of the background current ($I_{BG}$) was calculated by subtracting the $I_m$ in 144 NMDG solution ($I_{NMDG}$) from that in 144 Na solution ($I_{Na}$) (*Figure 1B*). On average, the $I_{NMDG}$ (7 pA) in wild-type OHCs was larger than that (4 pA) in *Tmc1*-knockout OHCs, but both were small (*Figure 1D*); the $I_{BG}$ in wild-type OHCs was 71 pA, while it was reduced to 18 pA in *Tmc1*-knockout OHCs (*Figure 1E*). Furthermore, the voltage dependence of $I_m$ and $I_{NMDG}$ was analyzed by applying a series of voltage-pulse stimuli to OHCs (*Figure 1F–I*). The $I_m$-V curves obtained from these measurements verified a reduced $I_m$ (*Figure 1G*) and a more negative reversal potential (*Figure 1H*) in *Tmc1*-knockout OHCs. After subtraction (only inward $I_{BG}$ was calculated because NMDG was applied extracellularly), it was clear that the $I_{BG}$ altered almost linearly with holding potentials and was dramatically reduced in *Tmc1*-knockout OHCs (*Figure 1I*).

We next considered whether overexpression of TMC1 would enhance the background current in hair cells. Three constructs were used for these experiments: enhanced green fluorescent protein (EGFP) as control, wild-type TMC1 (TMC1_WT), and TMC1 deafness (TMC1_dn) that carries a deletion mutation linked to deafness. Using cochlear injectoporation (*Xiong et al., 2014*), these constructs were delivered into wild-type OHCs on postnatal day 3 (P3). The cells were cultured for 1 day in vitro (1DIV) and then analyzed by immunostaining (*Figure 2A*) and voltage-clamp recording (*Figure 2B*). As revealed by HA antibody, exogenously expressed TMC1 was largely distributed

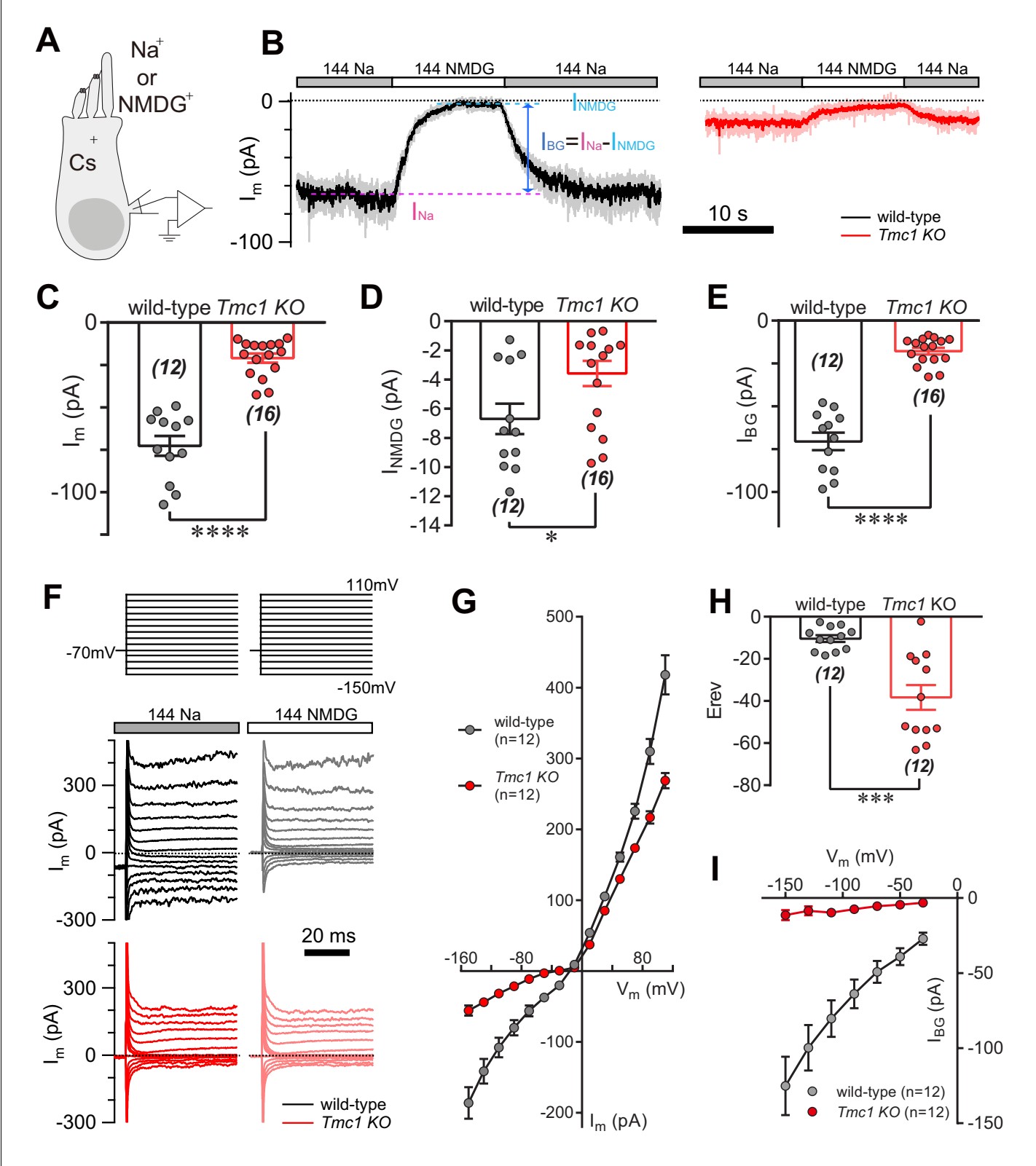

**Figure 1.** TMC1 mediates a background current in outer hair cells. (**A**) Diagram of the recording configuration. The P6 outer hair cells (OHCs, mostly P6 apical-middle OHCs if not specified otherwise) in acutely dissociated cochlea were whole-cell voltage-clamped with $Cs^+$ in the recording electrode and perfused with either 144 Na or 144 NMDG external solutions. 144 Na, regular recording solution; 144 NMDG, $Na^+$ substituted with $NMDG^+$. (**B**) Representative traces of membrane current ($I_m$) in OHCs from wild-type and *Tmc1*-knockout (*Tmc1 KO*) mice. The light gray and pink traces were

*Figure 1 continued on next page*

*Figure 1 continued*

recorded traces that were low-pass filtered to less noisy traces shown in black and red (similar filtering applied in the following figures). $I_{BG}$ (background current) was calculated by subtraction of $I_m$ in 144 Na ($I_{Na}$) and $I_m$ in 144 NMDG ($I_{NMDG}$) to exclude technical leak. (C–E) Quantification of the $I_{Na}$ (C), $I_{NMDG}$ (D), and $I_{BG}$ (E) measured from recordings similar to (B). Wild-type $I_{Na}$, −73 ± 6 pA, *Tmc1*-knockout $I_{Na}$, −21 ± 3 pA; wild-type $I_{NMDG}$, −7 ± 1 pA, *Tmc1*-knockout $I_{NMDG}$, −4 ± 1 pA; wild-type $I_{BG}$, −71 ± 5 pA, *Tmc1*-knockout $I_{BG}$, −18 ± 2 pA (F). Example of $I_m$ in wild-type (black and gray) and *Tmc1*-knockout (red and pink) OHCs undergoing a series of membrane depolarization, with tissues bathed in 144 Na followed by 144 NMDG. (G–I) Composite data showing I-V curve (G), reversal potential (H), and $I_{BG}$ (I) measured and calculated from recordings similar to (F). (G) I-V curve from recordings in 144 Na. (H) Mean reversal potentials calculated from I-V curve recorded in OHCs in 144 Na. (I) $I_{BG}$-V curve after subtracting $I_{NMDG}$. Only inward current was measured because NMDG was only applied extracellularly. The external solution contained 1.3 mM $Ca^{2+}$. The holding potential was −70 mV. Data are presented as mean ± SEM. N values are shown in each panel. *p<0.05, **p<0.01, ***p<0.001, Student's t-test.

DOI: https://doi.org/10.7554/eLife.47441.002

The following source data is available for figure 1:

**Source data 1.** TMC1 mediates a background current in outer hair cells.
DOI: https://doi.org/10.7554/eLife.47441.003

in the soma of OHCs (*Figure 2A*), consistent with previous observations (*Kawashima et al., 2011*). Overexpression of the EGFP and Tmc1_dn did not affect the $I_{BG}$ (18 pA and 16 pA) (*Figure 2C,D*), while the $I_{BG}$ in OHCs overexpressing Tmc1_WT was increased nearly 2.5-fold (43 pA) (*Figure 2C,D*). These data indicated that TMC1 functionally contributes a background leak conductance in hair cells.

It has been suggested that TMC2 is closely coupled with TMC1 in MET function. *Tmc2* expression in the cochlea is highest between P1 and P3, then falls after P4 (*Kawashima et al., 2011*). Exogenously expressed TMC2 was visibly located in hair bundles of OHCs, as shown by HA tag (*Figure 2—figure supplement 1*). We further examined the extent to which TMC2 could contribute a background current. Our data showed that the $I_{BG}$ was not altered in *Tmc2*-knockout OHCs at P1, P3, and P6 compared to controls (*Figure 2E,F*). Consistently, overexpression of TMC2 did not noticeably change the $I_m$ baseline (data not shown). In parallel, we analyzed the $I_{BG}$ in *Lhfpl5*-knockout OHCs. Interestingly, similar to *Tmc1*-knockout, there was no evident $I_{BG}$ in *Lhfpl5*-knockout OHCs (*Figure 2E,F*), consistent with the previous findings that LHFPL5 and TMC1 function in a common pathway (*Beurg et al., 2015b*; *Xiong et al., 2012*).

## TMC1-mediated leak current is not carried by the resting open MET channel

Because of existing tension in the hair bundle, the open probability of MET channels at rest in hair cells is significant (*Assad and Corey, 1992*; *Corey and Hudspeth, 1983*; *Johnson et al., 2012*). Thus, the $I_{BG}$ may come from the resting MET current. To determine the relationship between $I_{BG}$ and resting MET current, we analyzed the leak current during mechanical closure of hair bundles or in the presence of the MET channel blocker dihydrostreptomycin (DHS) (*Figure 3A*). As conductance through the MET channel is enhanced when the external $Ca^{2+}$ concentration is low, we carried out the experiments in 0.3 mM $Ca^{2+}$ to increase the readout of the leak current. A sinusoidal fluid jet deflected the hair bundle back and forth to open and close the MET channels (*Figure 3A*, inset). The $I_m$ was 98 pA at rest (*Figure 3A,B*), while the $I_m$ was 45 pA at fluid-jet-closed status (*Figure 3B*, #1). When OHC was perfused with solution containing 144 mM Na and 100 µM DHS (144 Na + 0.1 DHS), the current ($I_{DHS}$) was 55 pA (*Figure 3B*, #2 and #3). Moreover, the $I_{NMDG}$ was near zero when switching to 144 NMDG solution (*Figure 3B*, #3), from which $I_{Leak}$ was defined as MET-independent leak current by subtracting $I_{DHS}$ from $I_{NMDG}$. Thus, the $I_{Leak}$ persisted in either mechanical closure or pharmacological blockade of the MET channel (*Figure 3C*). We further examined the proportion of $I_{Leak}$ in $I_{BG}$ at different $Ca^{2+}$ concentrations, which became a major part of $I_{BG}$ when $[Ca^{2+}]_o$ was 1.3 mM and larger (*Figure 3D*). In general, the $I_{Leak}$ in 1.3 mM $[Ca^{2+}]_o$ was −51 pA in P6 wild-type OHCs and −17 pA in P6 *Tmc1*-knockout OHCs (*Figure 3E*), by subtracting currents recorded in 144 NMDG and 144 Na + 0.1 DHS. In the following experiments, we present the $I_{Leak}$ in 1.3 mM $[Ca^{2+}]_o$ as most of the measured membrane currents.

We queried where the $I_{Leak}$ comes from, hair bundle or soma. This question was examined by whole-cell voltage-clamp recordings in wild-type OHCs after removing the hair bundles (*Figure 3—figure supplement 1*). Compared to the $I_{Leak}$ recorded from hair-bundle preserved OHCs (48 pA),

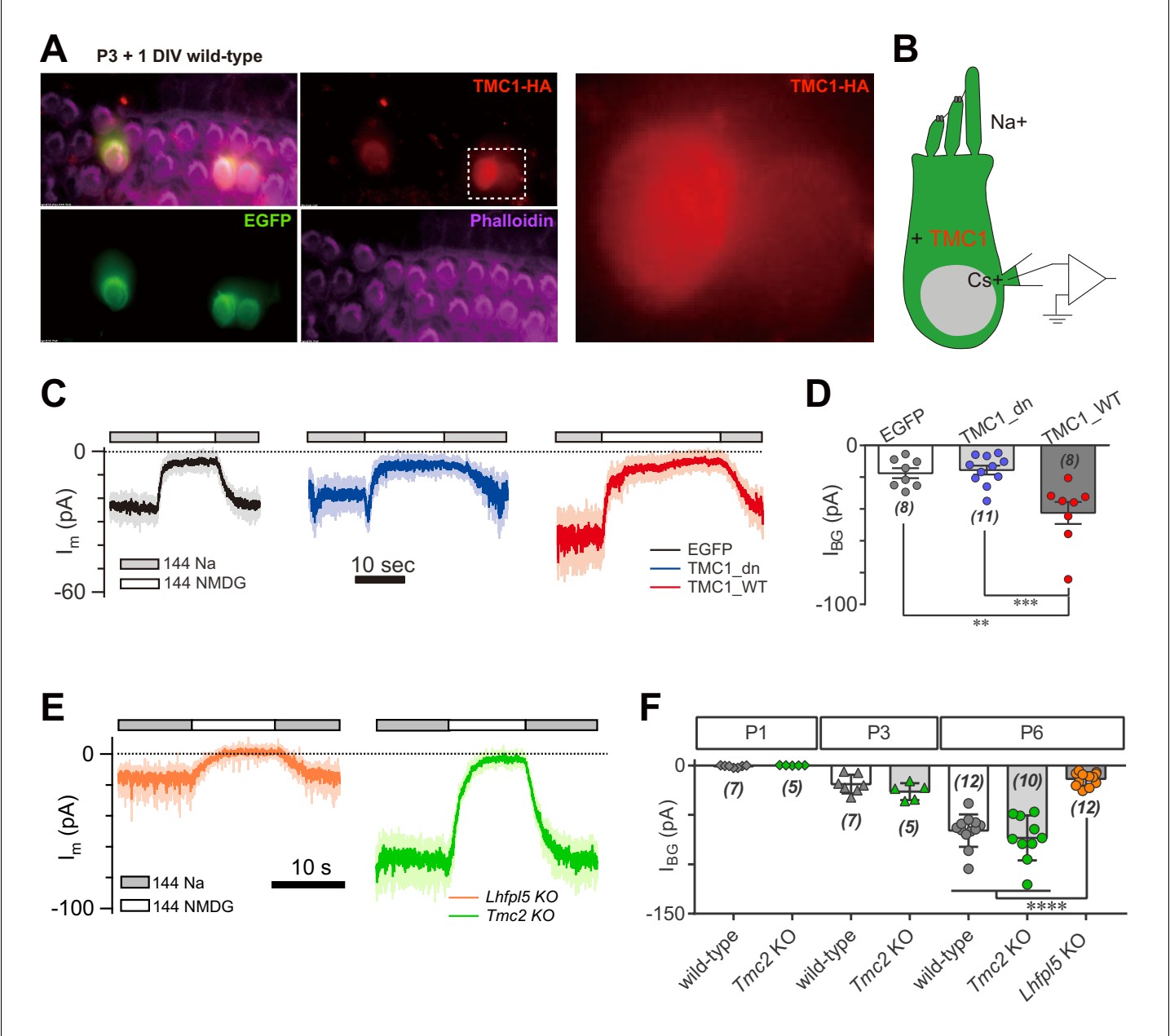

**Figure 2.** TMC1 but not TMC2 conducts the background current. (A) Exogenous expression of TMC1 in wild-type OHCs from organotypic P3 cochlear tissue cultured for 1 day in vitro (P3 + 1DIV). EGFP was co-expressed as an indicator. The OHCs were stained to show spatial distribution of TMC1 (recognized by HA antibody, red), EGFP (by GFP antibody, green), and actin-enriched stereocilia (by Phalloidin, magenta), with two OHCs from the white dashed frame shown in detail. (B) Diagram of the recording configuration. The OHCs expressed engineered TMC1 with EGFP and whole-cell voltage clamped with $Cs^+$ in the recording electrode and $Na^+$ extracellularly. (C) Examples of $I_m$ of wild-type OHCs at P3 + 1DIV, expressing control (EGFP), deafness TMC1 (TMC1_dn), or wild-type TMC1 (TMC1_WT). (D) Quantification of $I_{BG}$ from wild-type OHCs expressing EGFP, TMC1_dn, and TMC1_WT under conditions similar to those in (C). $I_{BG}$ values: EGFP, –17 ± 3 pA; TMC1_dn, –16 ± 3 pA; TMC1_WT, –43 ± 7 pA. (E) Representative traces of $I_{BG}$ in P6 *Tmc2*- and *Lhfpl5*-knockout OHCs from acutely dissociated cochleae. (F) Quantification of $I_{BG}$ measured from recordings similar to (E) from *Tmc2*- and *Lhfpl5*-knockout mice at assigned ages. $I_{BG}$ values: P1 wild-type, –1 ± 0 pA, P1 *Tmc2*-knockout, –0 ± 0 pA; P3 wild-type, –19 ± 4 pA, P3 *Tmc2*-knockout, –26 ± 4 pA; P6 wild-type, –66 ± 5 pA, P6 *Tmc2*-knockout, –73 ± 7 pA, P6 *Lhfpl5*-knockout, –14 ± 2 pA. The external solution contained 1.3 mM $Ca^{2+}$. The holding potential was –70 mV. Data are presented as mean ± SEM. N values are shown in each panel. *p<0.05, **p<0.01, ***p<0.001, one-way ANOVA.

DOI: https://doi.org/10.7554/eLife.47441.004

The following source data and figure supplement are available for figure 2:

**Source data 1.** TMC1 but not TMC2 conducts the background current.

*Figure 2 continued on next page*

*Figure 2 continued*

DOI: https://doi.org/10.7554/eLife.47441.006

**Figure supplement 1.** Localization of ectopically expressed TMC2 in OHCs.

DOI: https://doi.org/10.7554/eLife.47441.005

the $I_{Leak}$ from hair-bundle removed OHCs was much smaller (10 pA). This result suggests that the leak channels mostly function in hair bundles.

## Amino-acid substitutions in TMC1 alter the TMC1-mediated leak current

We next addressed whether essential amino acids of TMC1 affect the leak channel. It has been reported that six amino acids in TMC1 are critical for MET channel function by affecting the pore properties of the channel (*Pan et al., 2018*) (*Figure 4A*). We replaced these six amino acids with cysteine, as reported by *Pan et al. (2018)*, and expressed the mutations in *Tmc1*-knockout OHCs by injectoporation to assess the effects on the leak current (*Figure 4B*). As controls, we used TMC1_WT and TMC1_dn, and found that the $I_{Leak}$ in *Tmc1*-knockout OHCs at P3+1DIV was restored by TMC1_WT but not by TMC1_dn (*Figure 4C,D*). Among the cysteine-substituted TMC1 constructs, five out of the six amino acids failed to restore the leak current. Especially the G411C, N447C, D528C, and D569C mutations nearly abolished the $I_{Leak}$, while T532C partially restored it. Surprisingly, M412C, which has been linked to deafness in Beethoven mice (*Vreugde et al., 2002*), behaved like wild-type TMC1.

Treatment with MTSET (2-(trimethylammonium)ethyl methanethiosulfonate, bromide) did not, however, change the current baseline in OHCs when expressing any of the six cysteine-substituted TMC1 constructs (*Figure 4—figure supplement 1A*). This was not because of the insensitivity of cysteine, or a weak MTSET effect, because MTSET treatment did change the MET current amplitude in *Tmc1;Tmc2* double-knockout OHCs expressing M412C (*Figure 4—figure supplement 1B*) as previously reported (*Pan et al., 2018*). The cysteine replacement did not show a consistent pattern of modulation of the leak current or the MET current (*Figure 4—figure supplement 1C*), implying that different molecular mechanisms underlie the two types of current.

## Pharmacological blockade of the TMC1-mediated leak conductance

Next, we set out to evaluate the properties of the leak current by further analyzing its response to pharmacological inhibitors of the MET channel. We first examined the inhibitory effects of the commonly used MET channel blockers DHS, d-tubocurarine (dTC), and amiloride (*Figure 5A–D*). DHS had no blocking effect on the current baseline at a working concentration (100 µM) that blocks MET channels (*Figure 5A,B*). However, the background conductance was 50% inhibited at 487 µM DHS from the fit, 30-times the $IC_{50}$ of the MET channel (*Figure 5A,B*), and dTC and amiloride also affected the leak current, albeit at higher concentrations than the MET current (*Figure 5C,D*).

It has been reported that trivalent cations such as $Gd^{3+}$ and $La^{3+}$, block MET channels (*Farris et al., 2004*; *Kimitsuki et al., 1996*), so we applied $Gd^{3+}$ and $La^{3+}$ at various concentrations and monitored the inhibitory effects on evoked MET current as well as leak current (*Figure 5E–H*). Surprisingly, the leak current was not affected even when $[Gd^{3+}]_o$ reached 80 µM, the $IC_{50}$ for blocking the MET current (*Figure 5E,F*). However, the leak current was inhibited by $[Gd^{3+}]_o$ with an $IC_{50}$ of 541 µM (*Figure 5E,F*). Similarly, $[La^{3+}]_o$ inhibited the MET channel with an $IC_{50}$ of 259 µM and the leak current with an $IC_{50}$ of 531 µM (*Figure 5G,H*). It is noteworthy that the $I_{BG}$ included the $I_{Resting-MET}$ and the $I_{Leak}$ when the concentration of the blockers was low, but the $I_{BG}$ was mainly composed of the $I_{Leak}$ when the concentration of blockers was high enough (*Figure 5*).

## Ionic permeability of the TMC1-mediated leak conductance

To further characterize the leak current in OHCs, we carried out a series of ion-permeation tests using the cations $Li^+$, $Cs^+$, $Ba^{2+}$, $Zn^{2+}$, $Co^{2+}$, $Mg^{2+}$, and $Ca^{2+}$ (*Figure 6A,B*). Most of the cations shared a size of $I_{Leak}$ similar to $Na^+$, except for $Cs^+$ and $Ca^{2+}$ (*Figure 6A,B*). The $Cs^+$-conducted $I_{Leak}$ was slightly larger (*Figure 6A*), while 75 mM $Ca^{2+}$ robustly blocked the $I_{Leak}$ (*Figure 6B*). The $Ca^{2+}$ permeability of the leak channel was further determined from calculation of reversal potentials by a voltage ramp stimulation with $Ca^{2+}$ and DHS extracellularly and $Cs^+$ intracellularly (*Figure 6C*).

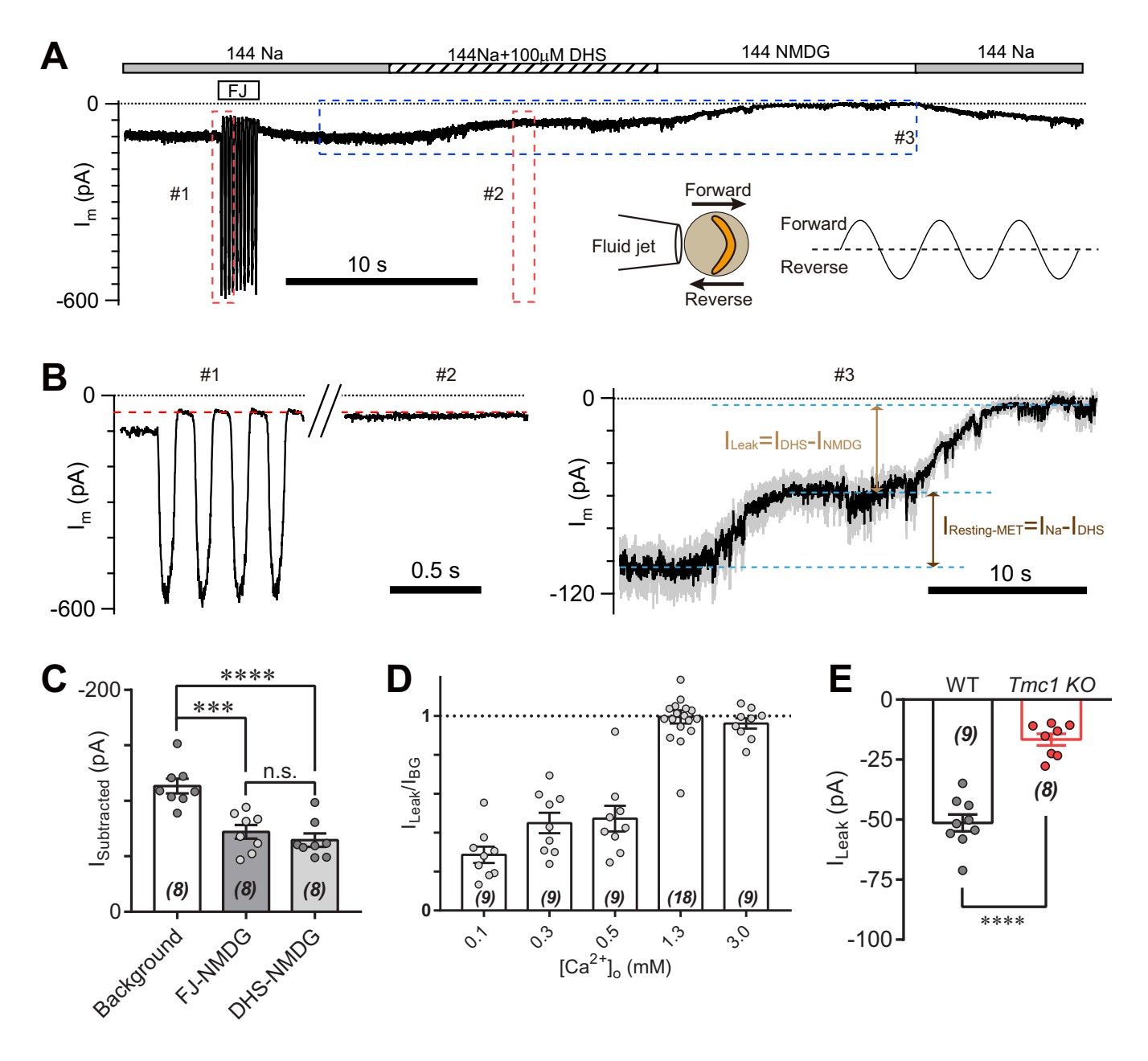

**Figure 3.** TMC1-mediated leak current is not carried by the resting open MET channel. (A) Representative $I_m$ trace showing fluid jet (FJ)-induced open and closed status of MET current and DHS-induced alteration of baseline current. The OHCs were bathed in external solution with 0.3 mM $Ca^{2+}$ instead of 1.3 mM $Ca^{2+}$. Insets: left, a diagram of fluid jet stimulation on a hair bundle; right, a 40 Hz sinusoidal stimulation protocol was used to induce forward and reverse deflection of the hair bundle. (B) Dashed frames #1, #2, and #3 in (A) are shown as enlarged traces. The baseline current was similar when the MET channels were closed by either FJ (#1) or DHS ($I_{DHS}$, #2), as highlighted with a red dashed line. As shown in #3, the DHS-sensitive resting MET current ($I_{Resting-MET}$) was calculated by subtraction of $I_{Na}$ and $I_{DHS}$. The baseline current was further closed by NMDG ($I_{NMDG}$). $I_{Leak}$ was defined as the subtraction of $I_{DHS}$ and $I_{NMDG}$. (C) Quantification of subtracted currents under different conditions: background, $-113 \pm 7$ pA; FJ-NMDG ($I_{Leak}$ subtracted from current baseline closed at negative FJ), $-72 \pm 6$ pA; DHS-NMDG ($I_{Leak}$ subtracted from that closed by 100 μM DHS), $-65 \pm 6$ pA. (D) Quantification of ratio of $I_{Leak}$ to $I_{BG}$ ($I_{Leak}/I_{BG}$) under different $[Ca^{2+}]_o$ conditions: 0.1 mM, $0.29 \pm 0.04$; 0.3 mM, $0.45 \pm 0.05$; 0.5 mM, $0.47 \pm 0.07$; 1.3 mM, $1.00 \pm 0.03$; 3.0 mM, $0.96 \pm 0.03$. (E) Quantification of the $I_{Leak}$ of OHCs measured in 1.3 $[Ca^{2+}]_o$. Wild-type $I_{Leak}$, $-51 \pm 3$ pA, *Tmc1*-knockout $I_{Leak}$, $-17 \pm 2$ pA. The external solution contained variable $Ca^{2+}$ concentration as indicated. The holding potential was $-70$ mV. Data are represented as mean ± SEM. N values are shown in each panel. *p<0.05, **p<0.01, ***p<0.001, (C) ANOVA; (E) Student's t-test.

DOI: https://doi.org/10.7554/eLife.47441.007

*Figure 3 continued on next page*

*Figure 3 continued*

The following source data and figure supplements are available for figure 3:

**Source data 1.** TMC1-mediated leak current is not carried by the resting open MET channel.
DOI: https://doi.org/10.7554/eLife.47441.009
**Figure supplement 1.** Removal of hair bundles disrupts leak current of OHCs.
DOI: https://doi.org/10.7554/eLife.47441.008
**Figure supplement 1—source data 1.** Removal of hair bundles disrupts leak current of OHCs.
DOI: https://doi.org/10.7554/eLife.47441.010

Compared to $Na^+$ and $Mg^{2+}$ permeability, the $Ca^{2+}$ likely provided an inhibition function on the leak channel (*Figure 6D*), which is different with the $Ca^{2+}$ permeability of the MET channel (around six for $P_{Ca}/P_{Cs}$ at P5 apical OHCs) (*Kim and Fettiplace, 2013*). Next, we monitored the background and MET currents in solutions containing different concentrations of $Ca^{2+}$ and $Na^+$. Results showed that the leak current was highly sensitive to $Ca^{2+}$; increasing when $[Ca^{2+}]_o$ declined, and inversely, decreasing when $[Ca^{2+}]_o$ escalated (*Figure 6E,F*). The MET current was initially reduced following the increase of $[Ca^{2+}]_o$, and reached a plateau after $[Ca^{2+}]_o$ was at least >10 mM (*Figure 6E,F*), by which the concentration was sufficient enough to block the leak current to an extent similar to that observed in *Tmc1*-knockout OHCs (*Figure 6G*).

## The leak current modulates action potential firing in IHCs

Next, we investigated the physiological relevance of the TMC1-mediated background conductance in auditory transduction. A significant leak conductance would be expected to depolarize the membrane and affect cell excitability. IHCs are innervated by the spiral ganglion neurons that transmit sound information to the CNS, and signal transmission from hair cells to the spiral ganglion; therefore possibly being affected by the leak conductance. Therefore, we measured the membrane potential ($V_m$) in IHCs bathed in external solution with 100 μM DHS by whole-cell current-clamp recording (*Figure 7A*). In wild-type IHCs, the resting $V_m$ varied (actively and periodically) in bursting and non-bursting states (*Figure 7A*). However, the resting $V_m$ was more hyperpolarized, and there was limited action potential firing in *Tmc1*-knockout IHCs (*Figure 7A*). With positive current injection, the *Tmc1*-knockout IHCs fired action potentials at threshold similar to wild-type IHCs (*Figure 7A*). Although the $V_m$ baseline in the non-bursting state was more hyperpolarized in wild-type IHCs, it was still more depolarized than the $V_m$ baseline in *Tmc1*-knockout IHCs (*Figure 7A,B*). This change of membrane excitability was also determined by monitoring the action potential bursting rate (*Figure 7C*) and the leak current (*Figure 7D*). We found that the leak current was smaller in IHCs than that in OHCs, which may be because of a different expression profile of potassium channels (*Marcotti et al., 2006*; *Marcotti et al., 2003*). Results also showed that, with ramp current injection, the firing threshold was similar, but the minimum injected current required to induce firing in *Tmc1*-knockout IHCs was ~20 pA greater than that in wild-type IHCs (*Figure 7E,G,H*). When depolarized by stepped current injection, the firing rate was lower in *Tmc1*-knockout IHCs and the rate-current curve was shifted to the right but finally reached a similar level when a larger current was injected (*Figure 7F,I*).

## The leak current follows the tonotopic gradient of the MET response in OHCs

The MET currents increase in OHCs from apex to base, which is considered as a manifestation of cochlear tonotopy. First, we examined the $I_{Leak}$ in OHCs along the cochlear coil (*Figure 8A*). As anticipated, we found a gradient in the leak current in wild-type OHCs, while the gradient was abolished in *Tmc1*-knockout OHCs (*Figure 8B,C*). We next analyzed the MET current along the cochlear coil when blocking the leak current with 35 mM $[Ca^{2+}]_o$, as 35 mM $[Ca^{2+}]_o$ was sufficient to block the leak current to an extent similar to TMC1 removal in OHCs (*Figure 6G*). Strikingly, the gradual increase in MET current amplitude was severely blunted in OHCs in the presence of 35 mM $[Ca^{2+}]_o$ (*Figure 8D,E*). Consistently, the OHCs lacking TMC1 lost the gradient of the MET currents as previously reported (*Beurg et al., 2014*) and we observed (*Figure 8D,E*). These data suggest that the tonotopic properties of the TMC1-participated leak channel and MET channel in OHCs could be

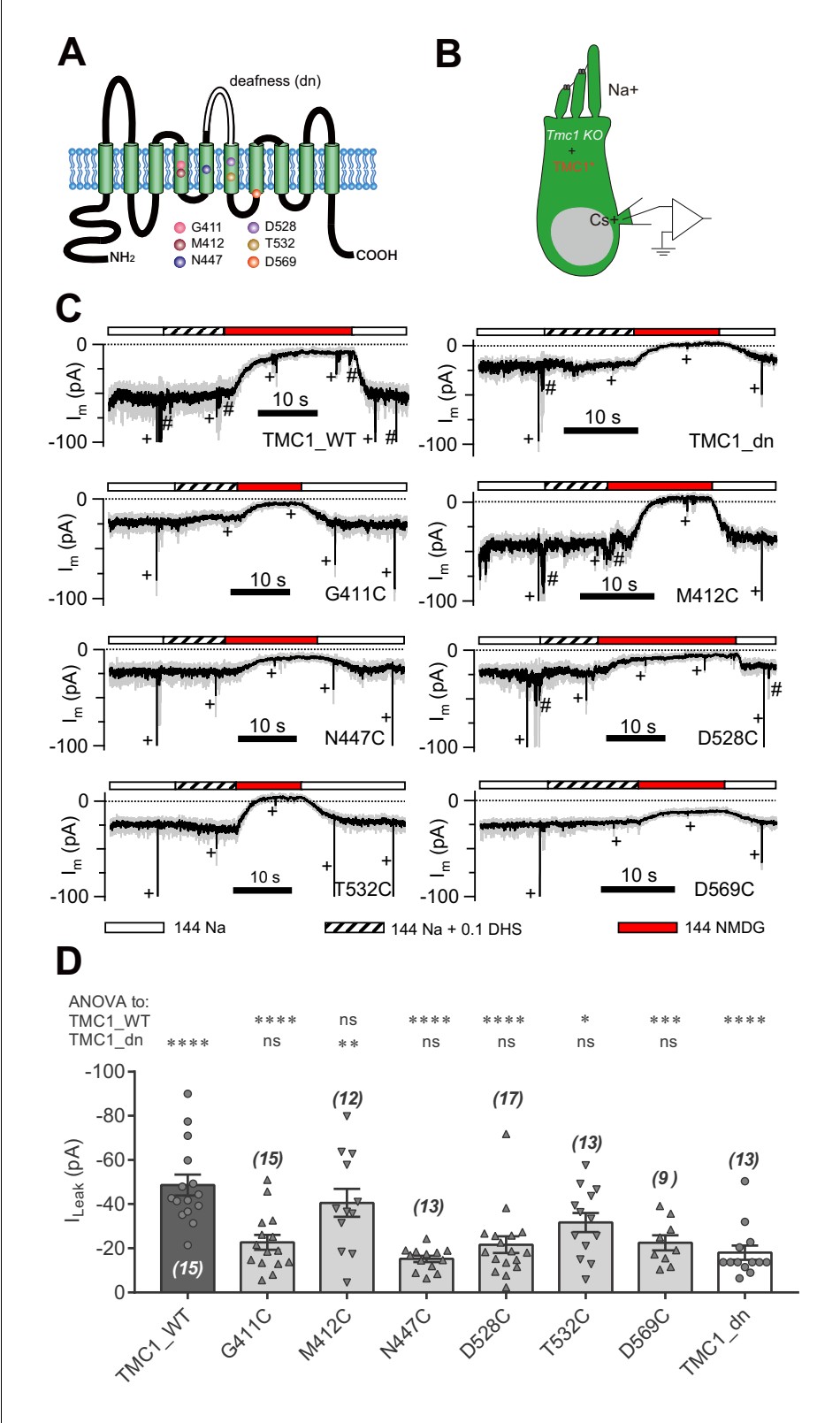

**Figure 4.** Amino-acid substitution in TMC1 alters the leak current. (A) TMC1 with 10 putative transmembrane domains. The six substituted amino acids are highlighted as colored balls in the predicted positions, and the deafness (dn) truncation is at the third extracellular loop between TM5 and TM6. (B) Diagram of the analysis of leak current in cultured *Tmc1*-knockout OHCs (P3 + 1 DIV) expressing modified TMC1 (TMC1*). (C) Representative traces showing the rescue of leak conductance in OHCs by control full-length TMC1 (TMC1_WT), deafness TMC1 (TMC1_dn), TMC1-G411C (G411C), TMC1-

*Figure 4 continued on next page*

*Figure 4 continued*

M412C (M412C), TMC1-N447C (N447C), TMC1-D528C (D528C), TMC1-T532C (T532C), and TMC1-D569C (D569C). Perfusion contents are indicated below. An 800 nm step deflection was applied to the hair bundle every 10 s by a glass probe. The glass probe induced MET currents are marked '+', accompanying unwanted MET currents and electrical artefacts induced by switching the perfusion system (#). Note that the MET current was truncated to better show the leak current. (D) Quantification of rescue by mTMC1 constructs. $I_{Leak}$ values: TMC1_WT, $-49 \pm 5$ pA, G411C, $-23 \pm 3$ pA; M412C, $-40 \pm 6$ pA, N447C, $-15 \pm 1$ pA; D528C, $-22 \pm 4$ pA, T532C, $-32 \pm 4$ pA, D569C, $-23 \pm 3$ pA, TMC1_dn, $-18 \pm 3$ pA. The rescue indexes of FL and dn were used to evaluate significant difference. Cell numbers are shown on each bar. The external solution contained 1.3 mM $Ca^{2+}$. The holding potential was $-70$ mV. Data are presented as mean $\pm$ SEM. *p<0.05, **p<0.01, ***p<0.001, ANOVA.

DOI: https://doi.org/10.7554/eLife.47441.011

The following source data and figure supplements are available for figure 4:

**Source data 1.** Amino-acid substitution in TMC1 alters the leak current.

DOI: https://doi.org/10.7554/eLife.47441.013

**Figure supplement 1.** Cysteine substitution in TMC1 affects the MET current and the leak current.

DOI: https://doi.org/10.7554/eLife.47441.012

**Figure supplement 1—source data 1.** Cysteine substitution in TMC1 affects the MET current and the leak current.

DOI: https://doi.org/10.7554/eLife.47441.014

modulated by external $Ca^{2+}$. The leak and MET current decreased from apex to base in *Tmc1*-knockout OHCs, which may correlate with the graded expression level of TMC1 along the cochlear coil. We further questioned temporal correlation of TMC1 and the leak conductance as *Tmc1* expression starts to increase from P3 and reaches a plateau in adult mice (*Kawashima et al., 2011*). We therefore investigated how the leak current changed in ageing OHCs before MET maturation (P3) and after the onset of hearing (P14). We found that the leak current amplitude increased from P3 to P6, and to P14 (*Figure 8—figure supplement 1*).

We next determined whether the change in macroscopic MET current represents a change in the unitary MET channel conductance, and whether the absence of the leak current disrupts the tonotopic gradient. The unitary MET channel analysis showed that 35 mM $[Ca^{2+}]_o$ reduced the unitary MET channel current to ~5 pA in both low-frequency and high-frequency OHCs (*Figure 9A,B*). These data further suggest that the extracellular $Ca^{2+}$ modulates leak conductance and MET channel properties, accordingly.

## Discussion

Our research demonstrated that in mouse hair cells, besides its function in MET, TMC1 is essential for a leak conductance (*Figures 1–3*). As shown by the mutagenesis experiments (*Figure 4*), at least four amino acids were critical for the leak conductance, as these constructs failed to restore the leak current after replacement of a single amino acid by cysteine. In addition, the leak conductance was inhibited by typical MET channel blockers, implying that TMC1 is the responsible component for the leak current (*Figure 5*). With TMC1 deficiency, the resting membrane potential was hyperpolarized, resulting in the absence of the spontaneous action potential firing in neonatal IHCs (*Figure 7*), and the removal of the leak conductance was coupled with abolishment of the gradient of MET conductances in OHCs (*Figures 8* and *9*). All these data pointed out a previously unappreciated non-MET role of TMC1 in mammals, by mediating a leak conductance and thereby participating tonotopy and regulating membrane excitability.

It has been recognized that leak conductance is generally used in the nervous system to regulate neuronal excitability and thus circuit activity; it recruits a variety of channels on the plasma membrane or endoplasmic reticulum (*Bers, 2014*; *Enyedi and Czirják, 2010*; *Lu et al., 2007*). In other species, TMC orthologues function in diverse ways according to their expression pattern in effector cells (*Guo et al., 2016*; *He et al., 2019*; *Wang et al., 2016*; *Yue et al., 2018*; *Zhang et al., 2015*; *Zhang et al., 2016*). Hence, our results strongly support the hypothesis that the excitability of cells and neural circuits that control processes from sensory transduction to motor function are commonly upregulated by TMC proteins in diverse organisms.

Of interest, the leak conductance differed from the MET conductance in several properties, although both are functional representations of TMC1. First, the leak current did not stem from the resting open MET channels (*Figure 3*). Second, the patterns of conductance change differed for the

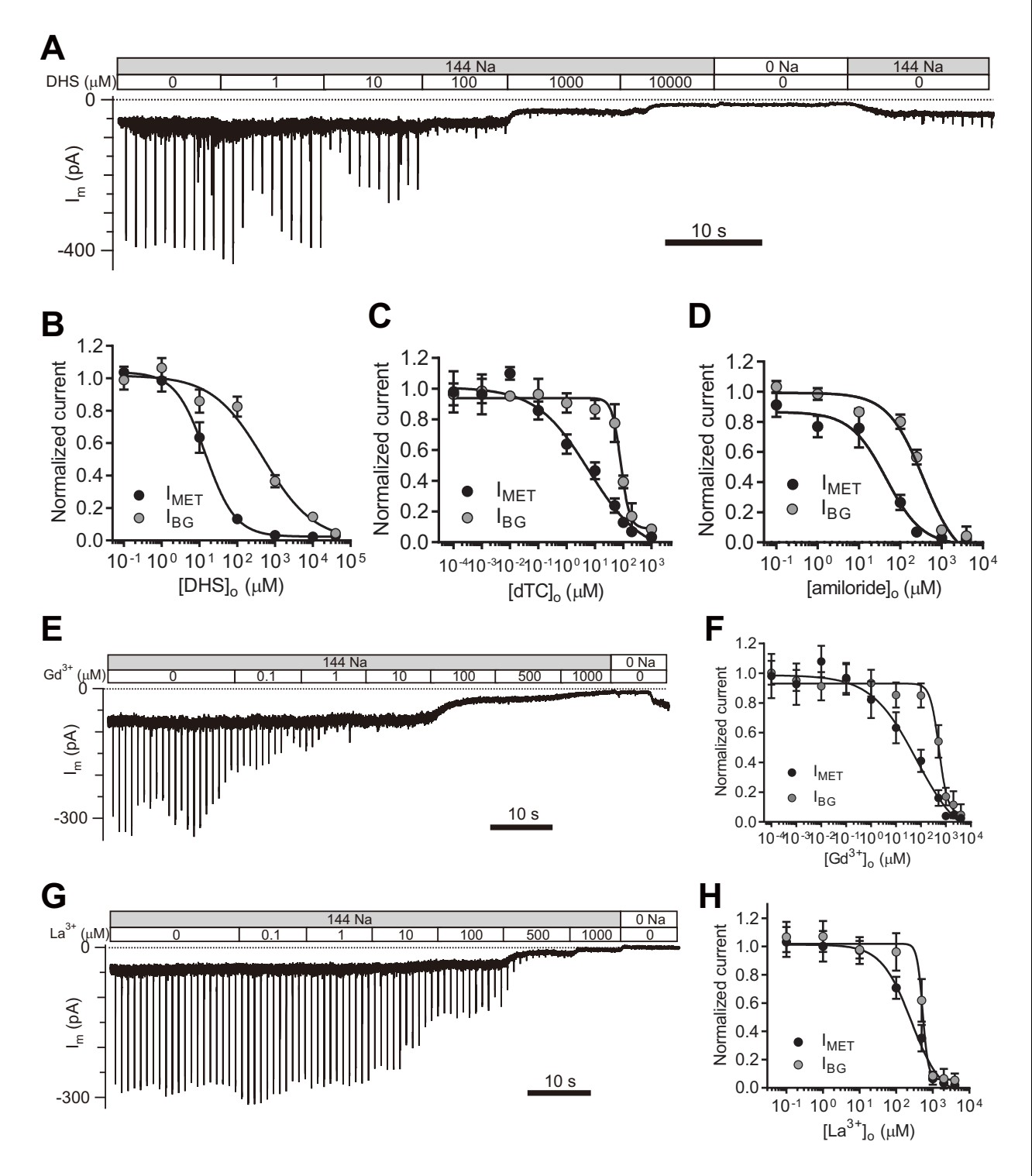

**Figure 5.** TMC1-mediated leak conductance is antagonized by MET channel blockers. (**A and B**) Representative trace (**A**) and statistical curve (**B**) of $I_m$ inhibition by DHS. A 10 Hz train of 800 nm step deflection was applied to the hair bundle by a glass probe to induce MET currents. $I_{MET}$ and $I_{BG}$ were calculated and plotted against the DHS concentration. As fitted, the $IC_{50}$ of DHS was 15 µM for the MET channels and 487 µM for the leak conductance. Cell numbers, 7–11. Hill slope: $I_{MET}$, −1.10; $I_{BG}$, −0.65. (**C and D**) Statistical dose curve of $I_m$ with graded concentrations of d-tubocurarine (dTC) (**C**) and amiloride (**D**). dTC $IC_{50}$: $I_{MET}$, 6 µM; $I_{BG}$, 82 µM. dTC Hill slope: $I_{MET}$, −0.47; $I_{BG}$, −2.80. dTC cell numbers, 5–15. Amiloride $IC_{50}$: $I_{MET}$, 46 µM; $I_{BG}$, 365 µM. Amiloride Hill slope: $I_{MET}$, −1.36; $I_{BG}$, −1.67. Amiloride cell numbers, 7–16. (**E and F**) Dosage effect of $Gd^{3+}$. Example trace (**E**) and statistical curve (**F**) of $I_m$ in OHCs during perfusion with solutions containing graded concentrations of $Gd^{3+}$. A train of 800 nm step deflection was

*Figure 5 continued on next page*

*Figure 5 continued*

applied to the hair bundle by a glass probe to induce MET currents. The MET and leak current amplitudes changed because of the channel sensitivity of $Gd^{3+}$ and NMDG. $IC_{50}$: $I_{MET}$, 66 µM; $I_{BG}$, 524 µM. Hill slope: $I_{MET}$, −0.48; $I_{BG}$, −2.49. Cell numbers, 7–16. (**G and H**) Dose effect of $La^{3+}$. Example trace (**G**) and dosage curve (**H**) of $I_m$ with $La^{3+}$ treatment. A train of 800 nm step deflection was applied to the hair bundle by a glass probe to induce MET currents. $IC_{50}$: $I_{MET}$, 259 µM; $I_{BG}$, 531 µM. Hill slope: $I_{MET}$, −1.06; $I_{BG}$, −5.67. Cell numbers, 7–8. For space reasons, 144 NMDG is shown as 0 Na. The external solution contained 1.3 mM $Ca^{2+}$. The holding potential was −70 mV. Data are presented as mean ± SEM.

DOI: https://doi.org/10.7554/eLife.47441.015

The following source data is available for figure 5:

**Source data 1.** TMC1-mediated leak conductance is antagonized by MET channel blockers.

DOI: https://doi.org/10.7554/eLife.47441.016

leak channel and the MET channel according to the amino-acid substitution experiment. Further, adding positive charge to these amino acids did not affect the leak conductance, as revealed by its insensitivity to treatment with MTSET (*Figure 4* and *Figure 4—figure supplement 1*), which was not identical to the effect on the MET conductance (*Pan et al., 2018*). An intriguing observation was that M412C did not disrupt the leak conductance with or without MTSET. Inversely though, previous evidence has shown that MTSET treatment of M412C reduced the MET current (*Pan et al., 2018*). In terms of the leak conductance, M412C behaved similarly to wild-type, which is not surprising, as studies have shown the distinct sensitivities of the leak and MET conductances to factors including channel blockers and calcium (*Figures 5* and *6*), and M412K causes deafness without affecting the MET current in Beethoven mice (*Marcotti et al., 2006*). Third, the leak channel shared a group of identical antagonists with the MET channel but had different blocking affinity (*Figure 5*). Both MET channel blockers, (*Figure 5A–D*) and non-selective cation channel blockers (*Figure 5E–H*) inhibited the leak current but with an $IC_{50}$ 5–10-fold higher than that for the MET channel. Last, extracellular high $Ca^{2+}$ blocked the leak conductance but not the MET channel (*Figure 6*). These lines of evidence indicate that TMC1 confers the leak conductance by a mechanism distinct from the MET channel.

The tonotopic gradient of conductance in OHCs is an important property of hair-cell MET (*Beurg et al., 2006*; *Ricci et al., 2003*; *Waguespack et al., 2007*). Interestingly, the TMC1-mediated leak conductance exhibits a tonotopic pattern in OHCs, in parallel with the tonotopicity of the MET current. The leak current still existed in *Tmc2*-knockout OHCs, but it was absent from *Tmc1*- or *Lhfpl5*-knockout OHCs (*Figure 2E,F*), which is consistent with the findings that the gradient of the MET response was lost in *Tmc1*- and *Lhfpl5*-knockout mice and preserved in *Tmc2*-deficient mice (*Beurg et al., 2015b*). The leak conductance and the MET conductance increased in wild-type OHCs but decreased in *Tmc1*-knockout OHCs along the cochlear coil (*Figure 8A,B*), which also coincides with the spatiotemporal *Tmc1* expression pattern (*Kawashima et al., 2011*) and the abundance of TMC1 proteins in graded numbers (*Beurg et al., 2018*). However, high $[Ca^{2+}]_o$ abolished both the leak current and the gradient of the MET response, defined by the analysis of the macroscopic MET current (*Figure 8*) and unitary MET current (*Figure 9*), which reflected different working mechanisms of the MET channel and the leak channel to $[Ca^{2+}]_o$. Our results showed that the leak conductance, together with the MET response, is tuned by extracellular $Ca^{2+}$ and other unknown determinants, which is not surprising as other factors, such as PIP2, also regulate MET channel pore properties (*Effertz et al., 2017*).

Because of limited information about the structure of TMC1, we do not yet know how TMC1 confers the leak conductance, at least it appears that the leak channel is located in the hair bundle (*Figure 3—figure supplement 1*). It was proposed that there are increasing numbers of TMC molecules (from 8 to 20) per MET site from low-frequency OHCs to high-frequency OHCs (*Beurg et al., 2018*). However, not all TMC1 proteins are localized at the MET site, where only up to two MET channel complexes exist (*Beurg et al., 2018*; *Beurg et al., 2006*; *Kurima et al., 2015*). Our data further describe a scenario that extra TMC1 proteins functionally couple with the leak channels with graded conductances in OHCs along the cochlear coil, coinciding with increased numbers of extra TMC1 molecules that are not at the MET site per hair bundle from apex to base (*Beurg et al., 2018*). Hence, we suggest a working model in which TMC1 functionally incorporates into a leak channel and the MET channel, rather than being the pore of the two channels, and tunes the activity of hair cells. However, this hypothesis needs to be further examined by structural and functional studies.

# Materials and methods

**Key resources table**

| Reagent type (species) or resource | Designation | Source or reference | Identifiers | Additional information |
|---|---|---|---|---|
| Gene (*Mus musculus*) | TMC1 | NCBI ID: 13409 | | |
| Gene (*Mus musculus*) | TMC2 | NCBI ID:192140 | | |
| Gene (*Mus musculus*) | Lhfpl5 | NCBI ID: 328789 | | |
| Strain, strain background (*Mus musculus*) | C57BL6 | Vitalriver | | |
| Genetic reagent (*Mus musculus*) | C57BL6 TMC1 knockout | MGI: J:184419 | Griffith AJ etc. | From JAX |
| Genetic reagent (*Mus musculus*) | C57BL6 TMC2 knockout | MGI: J:184419 | Griffith AJ etc. | From JAX |
| Genetic reagent (*Mus musculus*) | C57BL6 Lhfpl5 knockout | MGI: J:98396 | Johnson KR etc. | From JAX |
| Antibody | Chicken anti-GFP | aveslab | RRID:AB_10000240 | Cat:GFP-1020 (1:1000) |
| Antibody | Anti-mouse HA Clone 16B12 | Biolegend | RRID:AB_2565335 | Cat:901513 (1:500) |
| Antibody | Alexa FluroTM 488 goat anti-chicken IgG(H+L) | Invitrogen | RRID:AB_142924 | Cat: A-11039 Lot:1937504 (1:2000) |
| Antibody | Alexa FluroTM 568 goat anti-mouse IgG(H+L) | Invitrogen | RRID:AB_2534072 | Cat: A-11004 Lot:2014175 (1:1000) |
| Sequence-based reagent | TMC1-DF-F | Ruibio Tech | This paper | 5':tgagattaacaacaaggaat tcgtgcgtctcaccgttt |
| Sequence-based reagent | TMC1-DF-R | Ruibio Tech | This paper | 5':tgagacgcacgaattcctt gttgttaatctcatccatcaaggc |
| Sequence-based reagent | mTMC1-G411C-F | Ruibio Tech | This paper | 5': aatgtccctcctgTGTatgtt ctgtcccaccctgtttga |
| Sequence-based reagent | mTMC1-G411C-R | Ruibio Tech | This paper | 5':ACAcaggagggacattacc atgttcatttcattttttttcccacca |
| Sequence-based reagent | mTMC1-M412C-F | Ruibio Tech | This paper | 5':gtccctcctggggTGTttc tgtcccaccctgtttgactt |
| Sequence-based reagent | mTMC1-M412C-R | Ruibio Tech | This paper | 5':ACAccccaggagggacatt accatgttcatttcattttttttccca |
| Sequence-based reagent | mTMC1-N447C-F | Ruibio Tech | This paper | 5':tcttcttctaggcTGTtttg tatgtattcattctcgcctt |
| Sequence-based reagent | mTMC1-N447C-R | Ruibio Tech | This paper | 5':ACAgcctagaagaaga gcaaaaatgcgccccaggag |
| Sequence-based reagent | mTMC1-D528C-F | Ruibio Tech | This paper | 5':tctcaccgtttctTGTgtcct gaccacttacgtcacgat |
| Sequence-based reagent | mTMC1-D528C-R | Ruibio Tech | This paper | 5':ACAagaaacggtgagacgc acgaattcctgccccaccattgtttc |
| Sequence-based reagent | mTMC1-T532C-F | Ruibio Tech | This paper | 5':tgacgtcctgaccTGTta cgtcacgatcctcattggcga |
| Sequence-based reagent | mTMC1-T532C-R | Ruibio Tech | This paper | 5':ACAggtcaggacgtcaga aacggtgagacgcacgaattc |
| Sequence-based reagent | mTMC1-D569C-F | Ruibio Tech | This paper | 5':atacacagaattcTGT atcagtggcaacgtcctcgctct |

*Continued on next page*

*Continued*

| Reagent type (species) or resource | Designation | Source or reference | Identifiers | Additional information |
|---|---|---|---|---|
| Sequence-based reagent | mTMC1-D569C-R | Ruibio Tech | This paper | 5':ACAgaattctgtgtatgaag gatatccatattctaagtcccagca |
| Chemical compound, drug | Dihydrostreptomyc in sulfate | HarveyBio | | Cat: HZB1169-1 |
| Chemical compound, drug | d-Tubocurarine | TCI | | Cat: C0433 |
| Chemical compound, drug | Amiloride | Cayman | | Cat: 21069 |
| Chemical compound, drug | MTSET | Cayman | | Cat: 21069 |
| Chemical compound, drug | $GdCl_3$ | Sigma | | Cat: 439770–5G |
| Chemical compound, drug | $LaCl_3$ | Sigma | | Cat: 298182–10G |
| Chemical compound, drug | $CoCl_2$ | Sigma | | Cat: 60818–50G |
| Chemical compound, drug | $ZnCl_2$ | Sigma | | Cat: 793523–100G |
| Chemical compound, drug | $MgCl_2$ | Sigma | | Cat: M8266-100G |
| Chemical compound, drug | $CaCl_2$ | Sigma | | Cat: 746495–100G |
| Chemical compound, drug | CsCl | Sigma | | Cat:C3139-25G |
| Chemical compound, drug | KCl | Sigma | | Cat:P9333-500G |
| Chemical compound, drug | NaCl | Sigma | | Cat:S7653-1KG |
| Chemical compound, drug | NaOH | Sigma | | Cat:S8045-500G |
| Chemical compound, drug | KOH | Sigma | | Cat:306568–100G |
| Chemical compound, drug | CsOH | Sigma | | Cat:C8518-10G |
| Chemical compound, drug | BAPTA Tetrasodium salt hydrate | Bioruler | | Cat: RH100017-1g |
| Chemical compound, drug | EGTA | Sigma | | Cat: 03780 |
| Software, algorithm | Igor 6 | WaveMetrics, Inc | | |
| Software, algorithm | Micro-manager 1.4 | micro-manager.org | | |
| Software, algorithm | HEKA patchmaster | HEKA | | |
| Software, algorithm | Matlab 2014 | MathWorks | | |
| Software, algorithm | Prism GraphPad 6 | GraphPad Software. | | |
| Other | HEKA whole cell recording amplifier | HEKA | | Order Number: 895273 |
| Other | Micromanipulator | Sensapex | | Cat:uMp-3 |

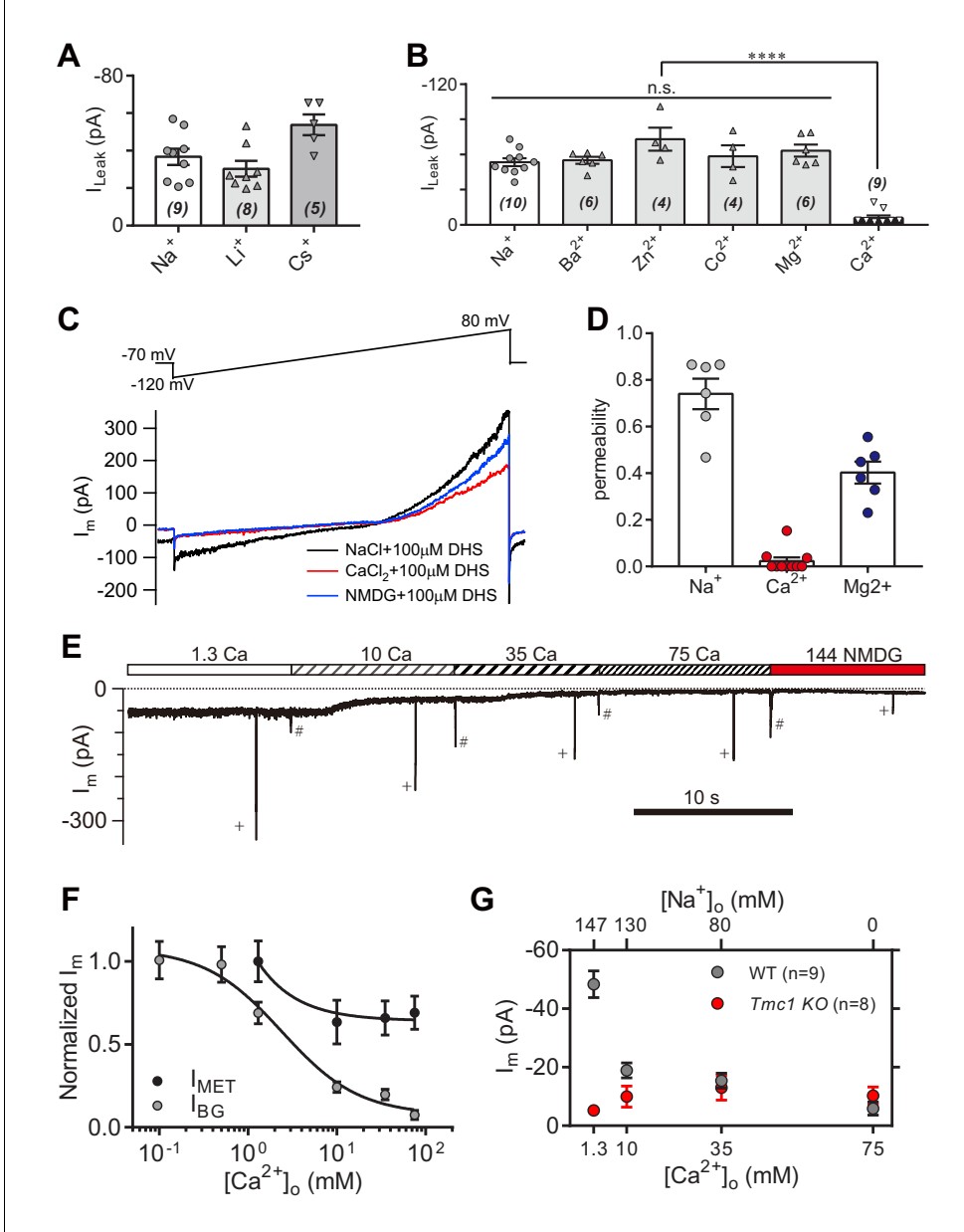

**Figure 6.** High-concentration $Ca^{2+}$ blocks the leak current but not MET current. (A) Monovalent cations $Li^+$ and $Cs^+$ conducted the leak current. In this experiment, 150 mM NaCl was substituted with 150 mM LiCl or 150 mM CsCl in the external solution. (B) Divalent cations 10 mM $Ba^{2+}$, 75 mM $Zn^{2+}$, 75 mM $Co^{2+}$, 150 mM $Mg^{2+}$, and 75 mM $Ca^{2+}$, conducted the leak current. The 150 mM NaCl was partially or completely replaced with cations as described in the Materials and methods. (C) Representative $I_m$ traces by ramp stimulation for calculation of ionic permeability of the leak channel. The extracellular ion was switched from 150 mM $Na^+$ to 75 mM $Ca^{2+}$, and to 150 $NMDG^+$, all containing 100 µM DHS. In the intracellular solution, 150 mM CsCl was used. (D) Quantification of ionic permeability calculated from similar recordings in (C). (E) Example trace of $I_m$ of OHCs during perfusion with solutions containing graded concentrations of $Ca^{2+}$ and $Na^+$. An 800 nm step deflection was applied to the hair bundle by a glass probe. The glass probe induced MET currents are marked '+', accompanying unwanted MET currents and artefacts induced by switching the perfusion system (#). (F) Dose curves of $I_{BG}$ and $I_{MET}$ in wild-type OHCs in different $Ca^{2+}$ and $Na^+$ concentrations (cell numbers, 9–20). (G) Quantification of dose-dependent background leak current in OHCs from wild-type (black) and *Tmc1*-knockout (red) mice when bathed in mixed $Ca^{2+}$ and $Na^+$. The ions and concentrations used in test external solutions were variable, as described in this figure legend and the Materials and methods. The holding potential was −70 mV. Data are presented as mean ± SEM. N values are shown in each panel. *p<0.05, **p<0.01, ***p<0.001, (B,D) ANOVA.
DOI: https://doi.org/10.7554/eLife.47441.017

The following source data is available for figure 6:

**Source data 1.** High-concentration $Ca^{2+}$ blocks the leak current but not MET current.
DOI: https://doi.org/10.7554/eLife.47441.018

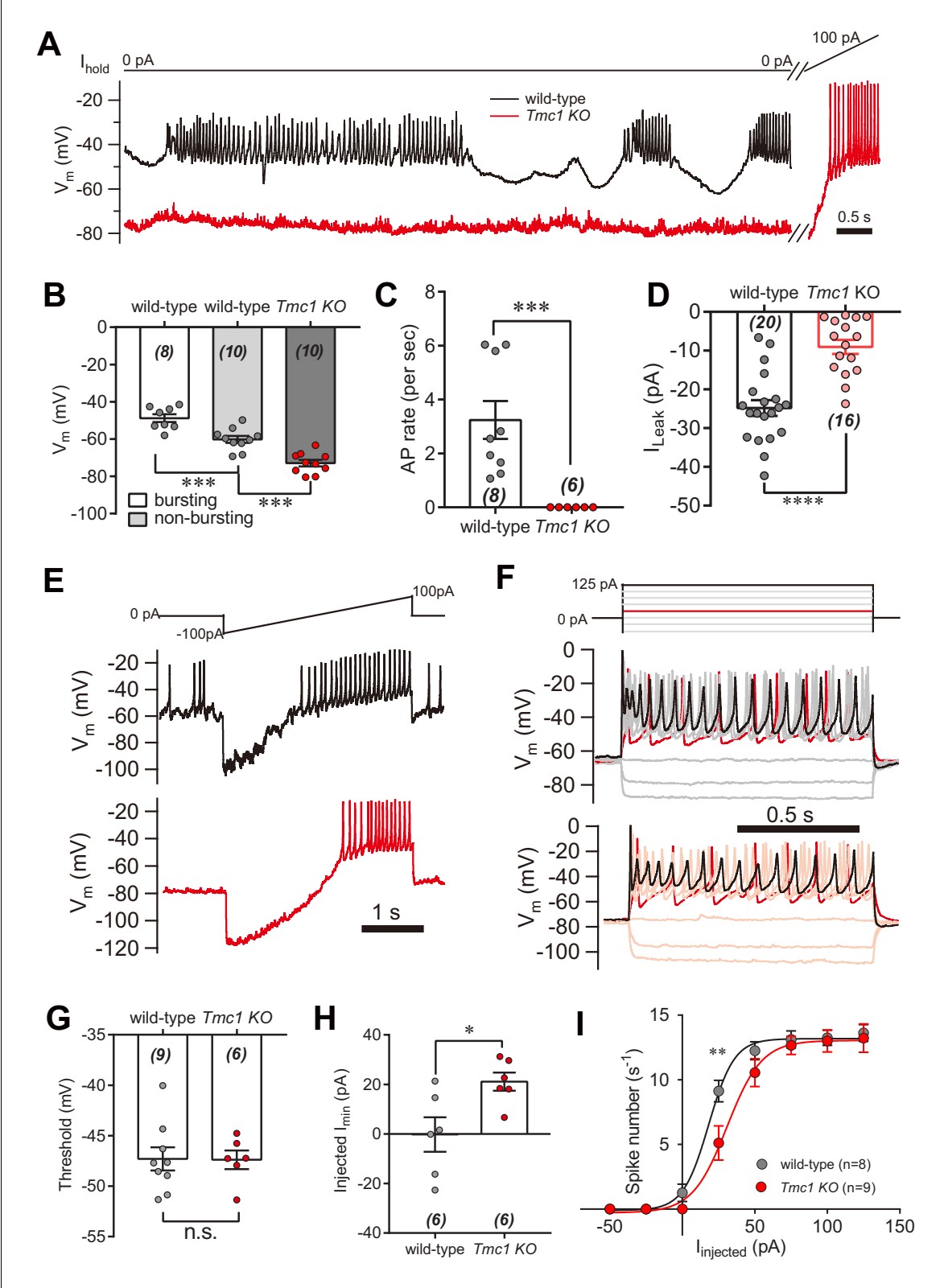

**Figure 7.** IHC excitability is down-regulated in *Tmc1*-knockout mice. (**A**) Representative current-clamp recording in IHCs bathed in external solution with 100 μM DHS from wild-type (black) and *Tmc1*-knockout (red) mice. For the most part, the IHCs were held at 0 pA. To define excitability, a ramp current was injected into the *Tmc1*-knockout IHCs to induce a burst of spikes. (**B**) Quantification of $V_m$ recorded in IHCs similar to (**A**). Values of $V_m$ in wild-type IHCs were defined as two states, bursting and non-bursting, which did not apply to *Tmc1*-knockout IHCs. $V_m$ of wild-type in bursting state,

*Figure 7 continued on next page*

*Figure 7 continued*

49 ± 2 mV; wild-type in non-bursting state, 60 ± 2 mV; *Tmc1*-knockout, 73 ± 2 mV. (C) Quantification of firing rate (spikes/s) in IHCs similar to (A). Values of firing rate: wild-type, 3.2 ± 0.7 Hz; *Tmc1*-knockout, 0 ± 0 Hz. (D) Quantification of $I_{Leak}$ from voltage-clamp recording in IHCs. Values of $I_{Leak}$: wild-type, 24 ± 4 pA; *Tmc1*-knockout, 14 ± 2 pA. (E) Representative current-clamp traces of $V_m$ in IHCs with ramp-current injection from –100 pA to +100 pA for 3 s. (F) Representative current-clamp recording in IHCs stimulated by a family of depolarization currents from –50 pA to +125 pA at 25 pA steps. (G) Quantification of firing threshold from data as in (E). Values of threshold were –47 ± 1 mV in wild-type OHCs and –47 ± 1 mV in *Tmc1*-knockout OHCs. (H) Quantification of minimum current injected (Injected $I_{min}$) to evoke an action potential from data as in (E). In wild-type OHCs: –0 ± 7 pA; in *Tmc1*-knockout OHCs: –21 ± 4 pA. (I) Quantification of numbers of spikes per second from data as in (F). Wild-type: 0 pA, 1.3 ± 0.7; 25 pA, 9.1 ± 0.8; 50 pA, 12.3 ± 0.7, 75 pA, 13.1 ± 0.7, 100 pA, 13.3 ± 0.6; 125 pA, 13.6 ± 0.7. *Tmc1*-knockout: 0 pA, 0 ± 0; 25 pA, 5.1 ± 1.3; 50 pA, 10.6 ± 1.1; 75 pA, 12.7 ± 0.7, 100 pA, 13.0 ± 0.9; 125 pA, 13.2 ± 1.1. In this figure, the external solution contained 1.3 mM $Ca^{2+}$ and 100 µM DHS. $K^+$ was used in the intracellular solution for current-clamp recordings in this figure except that $Cs^+$ was used for voltage-clamp recording in (D). Data are presented as mean ± SEM. N values are shown in each panel. *p<0.05, **p<0.01, ***p<0.001, (B) ANOVA; (C,D,G,H,I) Student's t-test.

DOI: https://doi.org/10.7554/eLife.47441.019

The following source data is available for figure 7:

**Source data 1.** IHC excitability is down-regulated in *Tmc1*-knockout mice.

DOI: https://doi.org/10.7554/eLife.47441.020

## Mouse strains and animal care

The mouse strains used in this study, B6.129-TMC1 < tm1.1Ajg>/J, B6.129-TMC2 < tm1.1Ajg>/J, and B6.129-Lhfpl5 < tm1Kjn>/Kjn, were from the Jackson Laboratory (Bar Harbor, ME). The experimental procedures on mice (Animal Protocol #: 15-XW1) were approved by the Institutional Animal Care and Use Committee of Tsinghua University.

## DNA constructs, cochlear culture, and injectoporation

Mouse *Tmc1* and *Tmc2* cDNAs were cloned into CMV-Script and pCDNA3.1- vectors, respectively. To obtain the *Tmc1*-deafness vector and amino-acid-substituted *Tmc1* constructs, specific primers were designed and used for PCR (*Supplementary File 1*). Cochlear culture and injectoporation were performed as previously described (*Xiong et al., 2014*). In brief, the organ of Corti was isolated from P3 mice and cut into three pieces in Dulbecco's modified Eagle's medium/F12 with 1.5 µg/ml ampicillin. For electroporation, a glass pipette (2 µm tip diameter) was used to deliver cDNA plasmids (0.2 µg/µl in 1 × Hanks' balanced salt solution) to hair cells in the sensory epithelium. EGFP was used as an indicator for the selection of transfected hair cells. A series of three pulses at 60 V lasting 15 ms at 1 s intervals was applied to cochlear tissues by an electroporator (ECM Gemini X2, BTX, CA). The cochlear tissues were cultured for 1 day in vitro and then used for electrophysiological recording.

## Electrophysiology

Hair cells were recorded using whole-cell voltage or current clamp as previously described (*Xiong et al., 2012*). All experiments were performed at room temperature (20–25˚C). Briefly, the basilar membrane with hair cells was acutely dissected from neonatal mice. The dissection solution contained (in mM): 141.7 NaCl, 5.36 KCl, 0.1 $CaCl_2$, 1 $MgCl_2$, 0.5 $MgSO_4$, 3.4 L-glutamine, 10 glucose, and 10 H-HEPES (pH 7.4). Then the basilar membrane was transferred into a recording chamber with recording solution containing (in mM): 144 NaCl, 0.7 $NaH_2PO_4$, 5.8 KCl, 1.3 $CaCl_2$, 0.9 $MgCl_2$, 5.6 glucose, and 10 H-HEPES (pH 7.4). For $I_{Leak}$ calculation, the cells were further bathed in recording solution containing 144 mM NMDG that replaced 144 mM NaCl. The acutely isolated or cultured basilar membrane was used for electrophysiological recording within 1 h. Hair cells were imaged under an upright microscope (BX51WI, Olympus, Tokyo, Japan) with a 60 × water immersion objective and an sCMOS camera (ORCA Flash4.0, Hamamatsu, Hamamatsu City, Japan) controlled by MicroManager 1.6 software (*Edelstein et al., 2010*). Patch pipettes were made from borosilicate glass capillaries (BF150-117-10, Sutter Instrument Co., Novato, CA) with a pipette puller (P-2000, Sutter) and polished on a microforge (MF-830, Narishige, Tokyo, Japan) to resistances of 4–6 MΩ. Intracellular solution contained (in mM): 140 CsCl, 1 $MgCl_2$, 0.1 EGTA, 2 Mg-ATP, 0.3 Na-GTP, and 10 H-HEPES, pH 7.2), except when CsCl was replaced with KCl in current-clamp. Hair cells were recorded with a patch-clamp amplifier (EPC 10 USB and Patchmaster software, HEKA Elektronik, Lambrecht/Pfalz, Germany). As measured, the liquid junction potential of the pipette with CsCl

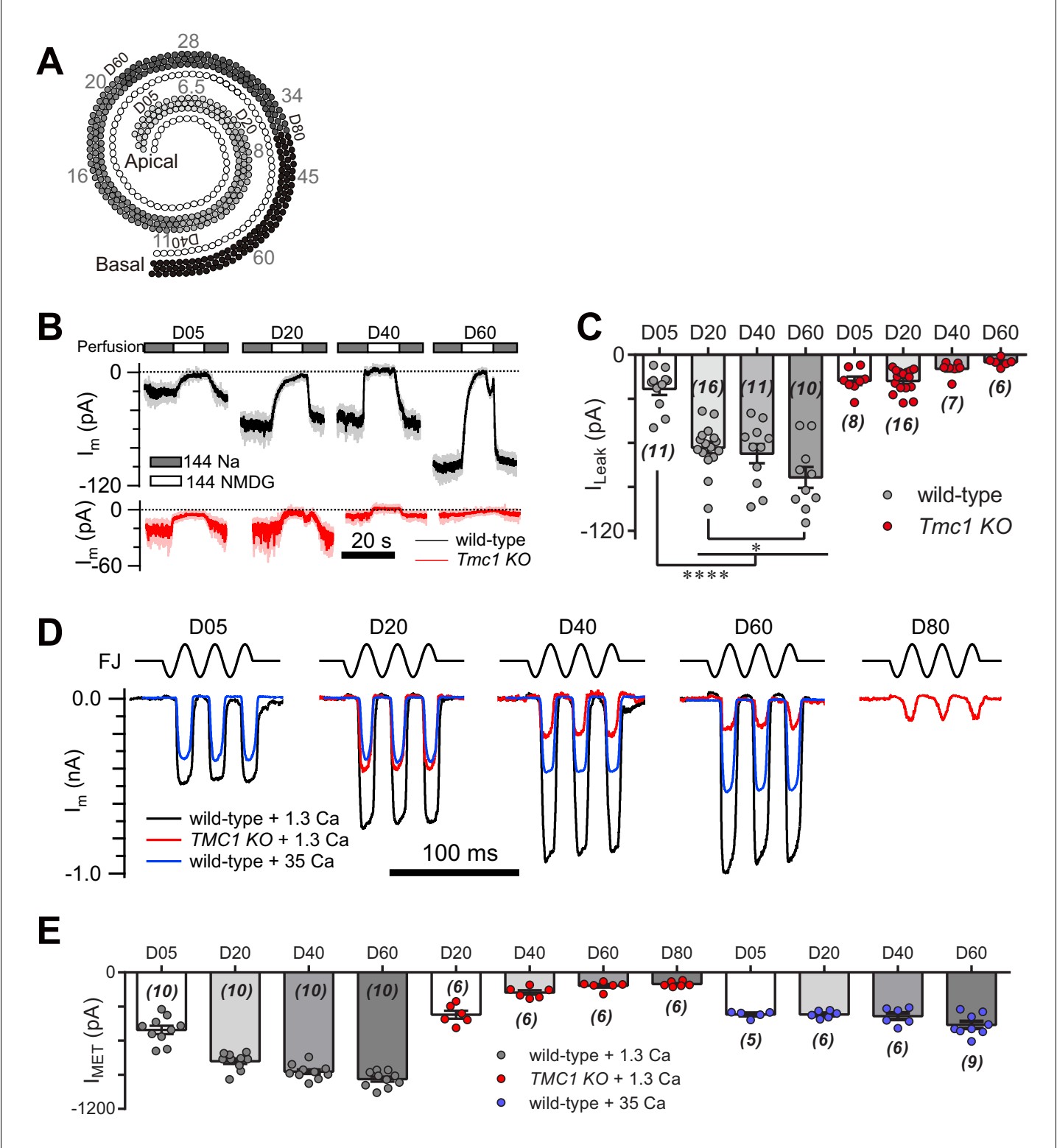

**Figure 8.** TMC1-mediated leak and MET currents in OHCs. (A) Diagram showing the tonotopic map in mouse hair cells (adapted from Figure 1B in *Kim and Fettiplace, 2013*), labeled with response frequencies (kHz, gray) and location (D% to apex, black). The apex and base are defined as 0 and 1, with reference to which D05, D20, D40, D60, and D80 represent distances of 0.05, 0.2, 0.4, 0.6, and 0.8. (B) Representative traces of $I_m$ recorded in OHCs at different locations along the cochlear coil, from wild-type (black) and *Tmc1*-knockout (red) mice. The external solution contained 1.3 mM $Ca^{2+}$. The apex and base are defined as 0 and 1, with reference to which D05, D20, D40, and D60 represent distances of 0.05, 0.2, 0.4, and 0.6. (C) *Figure 8 continued on next page*

*Figure 8 continued*

Quantification of location-specific $I_{Leak}$ from similar recordings to those in (B). Values of $I_{Leak}$ in wild-type OHCs (pA): D05, –23 ± 4; D20, –63 ± 4; D40, –67 ± 7; D60, –84 ± 7. $I_{Leak}$ values in *Tmc1*-knockout OHCs (pA): D05, –18 ± 3; D20, –18 ± 2; D40, –10 ± 2; D60, –5 ± 1. (D) Representative traces of location-specific MET current in wild-type OHCs when bathed in 1.3 mM or 35 mM $Ca^{2+}$ and *Tmc1*-knockout OHCs when bathed in 1.3 mM $Ca^{2+}$. A sinusoidal deflection was applied to the hair bundle by a fluid jet. (E) Quantification of location-specific macroscopic MET current. Values of $I_{MET}$ in wild-type OHCs in 1.3 mM $Ca^{2+}$ (pA): D05, –505 ± 37 pA; D20, –780 ± 24 pA; D40, –872 ± 21 pA; D80, –939 ± 22 pA. Values of $I_{MET}$ in wild-type OHCs in 35 mM $Ca^{2+}$ (pA): D05, –369 ± 13 pA; D20, –369 ± 13 pA; D40, –384 ± 30 pA; D60, –461 ± 31 pA. Values of $I_{MET}$ in *Tmc1*-knockout OHCs in 1.3 mM $Ca^{2+}$ (pA): D20, –371 ± 35 pA; D40, –177 ± 19 pA; D60, –117 ± 15 pA; D80, –102 ± 9 pA. The holding potential was –70 mV. In (C) and (E), data are presented as mean ± SEM with N values. *p<0.05, **p<0.01, ***p<0.001, ANOVA.

DOI: https://doi.org/10.7554/eLife.47441.021

The following source data and figure supplements are available for figure 8:

**Source data 1.** TMC1-mediated leak and MET currents in OHCs.

DOI: https://doi.org/10.7554/eLife.47441.023

**Figure supplement 1.** TMC1-dependent background leak current in ageing hair cells.

DOI: https://doi.org/10.7554/eLife.47441.022

**Figure supplement 1—source data 1.** TMC1-dependent background leak current in ageing hair cells.

DOI: https://doi.org/10.7554/eLife.47441.024

---

intracellular solution had a value of +4 mV in regular recording solution and –6 mV in 144 mM $NMDG^+$ solution, which was not corrected in the data shown.

For single-channel recordings, we followed published procedures (*Ricci et al., 2003*; *Xiong et al., 2012*). The intracellular solution was the same for macroscopic and microscopic current recording. To break tip-links, hair bundles were exposed to $Ca^{2+}$-free solution using a fluid jet (in mM): 144 NaCl, 0.7 $NaH_2PO_4$, 5.8 KCl, 5 EGTA, 0.9 $MgCl_2$, 5.6 glucose, and 10 H-HEPES, pH 7.4. After bundle destruction, fresh external solution was given to re-establish the corresponding extra-cellular ionic environment. Two external solutions were used: 3 mM $Ca^{2+}$ solution containing (in mM) 144 NaCl, 0.7 $NaH_2PO_4$, 5.8 KCl, 3 $CaCl_2$, 0.9 $MgCl_2$, 5.6 glucose, and 10 H-HEPES, pH 7.4; and 35 mM $Ca^{2+}$ solution containing (in mM) 80 NaCl, 0.7 $NaH_2PO_4$, 5.8 KCl, 35 $CaCl_2$, 0.9 $MgCl_2$, 5.6 glucose, and 10 H-HEPES, pH 7.4. Only traces with obvious single-channel events were included for analyzing.

The sampling rate was 1 kHz for leak current recording, 50 kHz for the IV protocol and current-clamp recording, and 100 kHz for unitary channel recording. The voltage-clamp used a –70 mV holding potential, and the current-clamp was held at 0 pA. Only recordings with a current baseline <20 pA in NMDG solution were used for statistical analysis.

## Hair bundle stimulation and removal

The hair bundle was deflected by two types of mechanical stimulus, fluid jet and glass probe. The fluid jet stimulation was performed as described previously (*Beurg et al., 2014*). In brief, a 40 Hz sinusoidal wave stimulus was delivered by a 27-mm-diameter piezoelectric disc driven by a home-made piezo amplifier pipette with a tip diameter of 3–5 µm positioned 5–10 µm from the hair bundle to evoke maximum MET currents. For glass probe stimulation, hair bundles were deflected with a glass pipette mounted on a P-885 piezoelectric stack actuator (Physik Instrumente, Karlsruhe, Germany). The actuator was driven with voltage steps that were low-pass filtered at 10 KHz. To avoid bundle damage caused by overstimulation, the glass probe was shaped to have a slightly smaller diameter than the hair bundles, and the stimulation distance was 800 nm for macroscopic current and 100 nm for unitary channel recording. For hair bundle removal, a pipette with 1 µm diameter tip was used to suck away the hair bundles of the target OHCs. The hair-bundle-removed OHCs with good condition were further recorded. Examined by a fluid-jet stimulation, the OHCs without obvious MET current were further measured for the leak current.

## Inhibitors, ion substitution, permeability, and perfusion

In *Figure 5*, DHS, dTC, amiloride, $GdCl_3$, and $LaCl_3$ were added as calculated to the recording solution (in mM) 144 NaCl, 0.7 $NaH_2PO_4$, 5.8 KCl, 1.3 $CaCl_2$, 0.9 $MgCl_2$, 5.6 glucose, and 10 H-HEPES (pH 7.4). Dose-inhibition curves were fitted with a Hill equation:

$$I_x/I_{max} = X^h/(K^h + X^h))$$

Where $K$ is the half-inhibition dose (IC50) and $h$ is the Hill slope. $I_{max}$ is the maximal current in control condition.

In *Figure 6*, all the ion substitution solutions were derived from a simplified external solution (in mM): 147 NaCl, 1.3 CaCl₂, 5.6 glucose, and 10 H-HEPES (pH 7.4). In *Figure 6A*, LiCl and CsCl were 150 mM, completely substituting for NaCl. In *Figure 6B*, the $Ba^{2+}$ solution was (in mM) 10 BaCl₂, 137 NaCl, 1.3 CaCl₂, 5.6 glucose, and 10 H-HEPES (pH 7.4); the $Zn^{2+}$ solution was 75 ZnCl₂, 75 NaCl, 1.3 CaCl₂, 5.6 glucose, and 10 H-HEPES (pH 7.4); the $Co^{2+}$ solution was 75CoCl₂, 75 NaCl, 1.3 CaCl₂, 5.6 glucose, and 10 H-HEPES (pH 7.4); the $Mg^{2+}$ solution was 150 MgCl₂, 5.6 glucose, and 10 H-HEPES (pH 7.4); and the $Ca^{2+}$ solution was 75 CaCl₂, 75 NaCl, 5.6 glucose, and 10 H-HEPES (pH 7.4).

$Ca^{2+}$ permeability was measured by performing whole-cell voltage-clamp recording on P6 OHCs, with intracellular solution containing (in mM): 140 CsCl, 1 MgCl₂, 0.1 EGTA, 2 Mg-ATP, 0.3 Na-GTP, and 10 H-HEPES, pH 7.2. A voltage-ramp stimulation from −120 to 80 mV lasting for 2 s was applied to calculate the reversal potential. For measuring $Na^+$ permeability, OHCs were perfused with the external solution containing (in mM): 150 NaCl, 1.3 CaCl₂, 5.6 glucose, and 10 H-HEPES. For

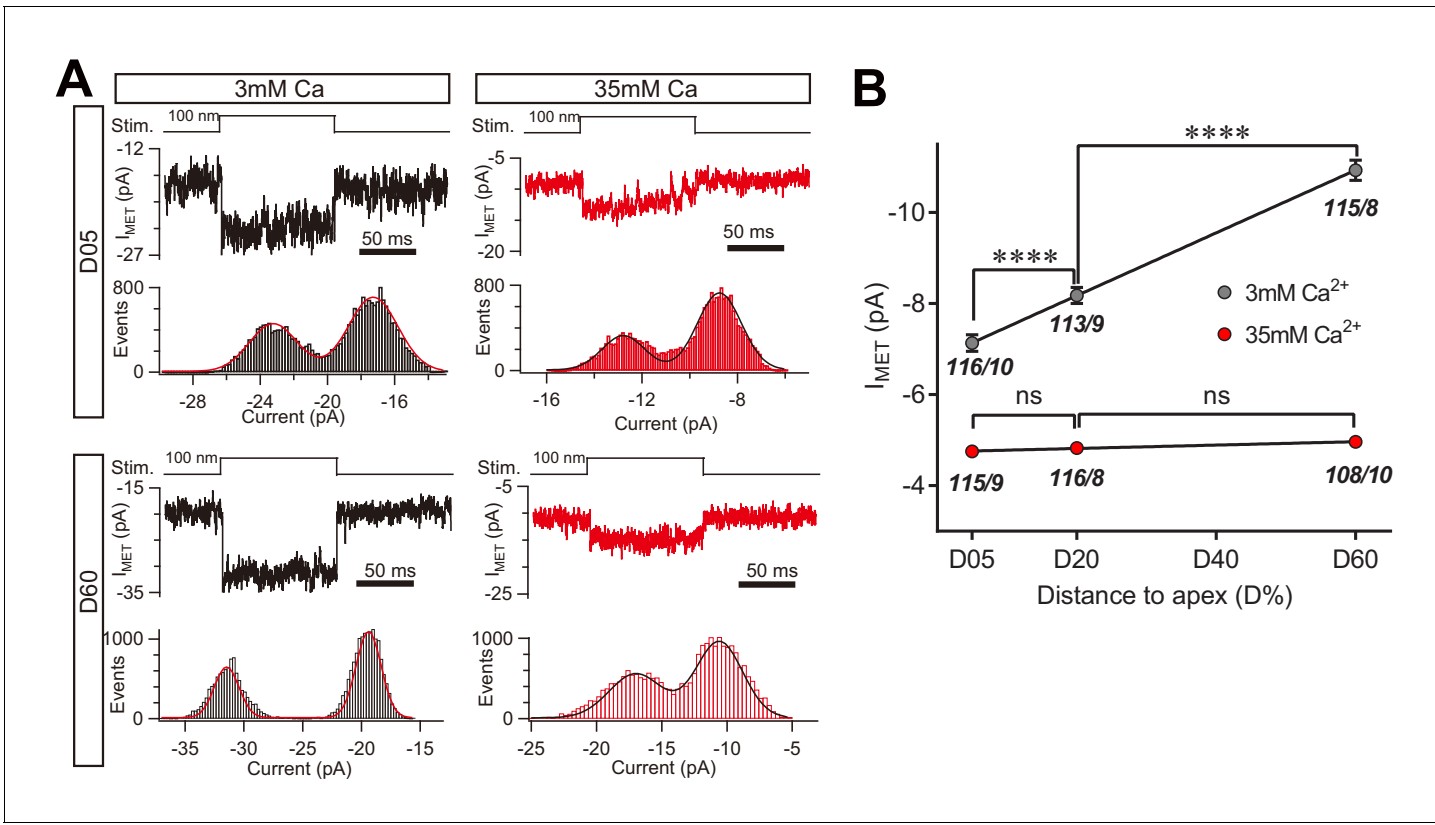

**Figure 9.** High $Ca^{2+}$ removes the MET conductance gradient as revealed by unitary channel analysis. (**A**) Location-specific single MET channel recording from wild-type OHCs in solution with 3 mM or 35 mM $Ca^{2+}$ at D05 or D60. The traces were chosen to show nice dual-peak fitting but did not represent normal flickers. A 100 nm step deflection was applied to the hair bundle by a glass probe. (**B**) Statistical analysis of location-specific unitary MET channel current. Values of unitary $I_{MET}$ in 3 mM $Ca^{2+}$: D05, −7.0 ± 0.2 pA; D20, −7.9 ± 0.2 pA; D60, −10.6 ± 0.2 pA. Values of $I_{MET}$ in 35 mM $Ca^{2+}$: D05, −4.7 ± 0.1 pA; D20, −4.8 ± 0.1 pA; D60, −4.9 ± 0.1 pA. The holding potential was −70 mV. N values are shown as events/cells. Data are presented as mean ± SEM. *p<0.05, **p<0.01, ***p<0.001, ANOVA.
DOI: https://doi.org/10.7554/eLife.47441.025

The following source data is available for figure 9:

**Source data 1.** High $Ca^{2+}$ removes the MET conductance gradient as revealed by unitary channel analysis.
DOI: https://doi.org/10.7554/eLife.47441.026

measurement of $Ca^{2+}$ or $Mg^{2+}$ permeability, 150 NaCl was substituted with 75 $Ca^{2+}$ or 75 $Mg^{2+}$ supplemented with 75 $NMDG^+$. To eliminate the influence of technical leak, an identical voltage-ramp stimulation was applied on each recorded OHC in 150 NMDG. The part of the inward current trace was fitted linearly to calculate the voltage value cross point between interest of ion and NMDG solution, which represented the reverse potential of the leak between this ion and $Cs^+$. The relative permeability of the monovalent cation was calculated as described (*Hille, 2001*)

$$P_X/P_{Cs} = [Cs]_i exp(Er_{rev}F/RT)/[X]_o$$

And for divalent cations, the equation was:

$$P_X/P_{Cs} = \gamma_{Cs}[Cs]_i exp(Er_{rev}F/RT)[exp(E_{rev}/RT)+1]/4\gamma_X[X]_o$$

For which $\gamma_{Cs}$ = 0.70 (*Hille, 2001*), $\gamma_{Ca}$ = 0.4657, $\gamma_{Mg}$ = 0.5271 (*Rodil and Vera, 2001*).

$E_{rev}$ means reversal potential, $F$ and $R$ mean Faraday constant and gas constant, $T$ means absolute temperature. For calculation, 25°C was used as the value for room temperature.

For the Ca-NMDG solution in *Figure 6E-F*, 1 $CaCl_2$ was exchanged for 2 NMDG-Cl. For the Na-Ca solution in *Figure 6G*, 2 NaCl was exchanged for 1 $CaCl_2$. The osmotic pressure of each solution was re-adjusted to 300–320 mOsm/kg with sucrose, and the pH was adjusted to 7.4.

The gravity perfusion system (ALA-VM8, ALA Scientific Instruments, Farmingdale, NY) is controlled manually to switch and deliver solutions. The perfusion tubing and tip were modified as previously reported (*Wu et al., 2005*). For cochlear tissue, the perfusion tip was placed 2–3 mm from the patched hair cell and the perfusion rate was ~0.5 ml/min. Extra solution in the recording dish was removed by a peristaltic pump (PeriStar, World Precision Instruments, Sarasota, FL) to maintain a steady liquid level.

## Data analysis

Every experiment contained at least three biological replicates, which were collected at least every 2 weeks within a 3 month window to maintain the stability of a data set. For certain experiments, such as single-channel recording, the trace numbers were over 100. All cell numbers were noted in the figure legends. Multiple recordings from one cell with the identical stimulus protocol were considered as technical replications, which were averaged to generate a single biological replication representing value/data from one cell. Data were managed and analyzed with Excel (Microsoft), Prism 6 (GraphPad Software, San Diego, CA), and Igor pro 6 (WaveMetrics, Lake Oswego, OR). The current traces similar to *Figure 1B* were low-pass filtered to less noisy traces with the smoothing function (Binomial 20) provided by Igor software. All data are shown as mean ± SEM. We used student's T-test for one-to-one comparison and ANOVA for multiple comparisons to determine statistical significance (*$p < 0.05$, **$p < 0.01$, ***$p < 0.001$). Values and N numbers are defined in the figures and figure legends.

## Acknowledgements

We thank Drs Ulrich Mueller, Bailong Xiao, Xin Liang, Wei Zhang, and members of Xiong laboratory for helpful discussions and critical proof-reading of this manuscript. This work was supported by the National Natural Science Foundation of China (31522025, 31571080, 81873703, and 3181101148), Beijing Municipal Science and Technology Commission (Z181100001518001), NSFC-RGC joint research scheme grant (N_HKUST614/18), and a startup fund from the Tsinghua University-Peking University Joint Center for Life Sciences.

## Additional information

### Funding

| Funder | Grant reference number | Author |
| --- | --- | --- |
| National Natural Science Foundation of China | 31522025 | Wei Xiong |

| Beijing Municipal Science & Technology Commission | Z181100001518001 | Wei Xiong |
| National Natural Science Foundation of China | 31571080 | Wei Xiong |
| National Natural Science Foundation of China | 81873703 | Wei Xiong |
| National Natural Science Foundation of China | 3181101148 | Wei Xiong |
| National Natural Science Foundation of China | Joint research scheme grant (N_HKUST614/18) | Pingbo Huang |
| Research Grants Council, University Grants Committee | Joint research scheme grant (N_HKUST614/18) | Wei Xiong |

The funders had no role in study design, data collection and interpretation, or the decision to submit the work for publication.

## Author contributions

Shuang Liu, Data curation, Software, Formal analysis, Investigation, Methodology; Shufeng Wang, Linzhi Zou, Investigation, Methodology; Jie Li, Chenmeng Song, Jiaofeng Chen, Qun Hu, Lian Liu, Investigation; Pingbo Huang, Funding acquisition, Visualization, Writing—review and editing; Wei Xiong, Conceptualization, Resources, Data curation, Software, Formal analysis, Supervision, Funding acquisition, Validation, Investigation, Visualization, Methodology, Writing—original draft, Project administration, Writing—review and editing

## Author ORCIDs

Pingbo Huang (iD) http://orcid.org/0000-0002-4560-8760
Wei Xiong (iD) https://orcid.org/0000-0002-2784-7696

## Ethics

Animal experimentation: The experimental procedures on mice were approved by the Institutional Animal Care and Use Committee of Tsinghua University (Animal Protocol #: 15 XW1).

## Decision letter and Author response

Decision letter https://doi.org/10.7554/eLife.47441.030
Author response https://doi.org/10.7554/eLife.47441.031

# Additional files

## Supplementary files

• Supplementary file 1. Primers used for generating desired truncation and mutations in mouse *Tmc1* cDNA. Specific primers were designed for PCR of the *Tmc1*-deafness vector and amino-acid-substituted *Tmc1* constructs, based on the pCDNA3.1 vector containing mouse *Tmc1* cDNA. DF, deafness; F, forward; R, reverse.
DOI: https://doi.org/10.7554/eLife.47441.027
• Transparent reporting form DOI: https://doi.org/10.7554/eLife.47441.028

## Data availability

All data generated or analysed during this study are included in the manuscript and supporting files.

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
