## [Decision Letter]

Thank you for submitting your article "TMC1 confers a leak conductance to modulate excitability of auditory hair cells in mammals" for consideration by *eLife*. Your article has been reviewed by three peer reviewers, one of whom is a member of our Board of Reviewing Editors, and the evaluation has been overseen by Richard Aldrich as the Senior Editor. The following individuals involved in review of your submission have agreed to reveal their identity: Peter G Barr-Gillespie (Reviewer #2); Anthony J Ricci (Reviewer #3).

The reviewers have discussed the reviews with one another and the Reviewing Editor has drafted this decision to help you prepare a revised submission.

Summary:

This is an interesting study which examined the contribution of TMC1 to the resting membrane "leak" conductance of hair cells in the mammalian cochlea. An understanding of the proteins involved in forming MET channels and other sources of resting membrane current in hair cells is lacking and continues to be an issue of great controversy. The authors approach this question by performing whole cell voltage clamp recordings from OHCs and IHCs in cochlea isolated from pre-hearing mice, in combination with pharmacological and genetic manipulations. The experiments provide evidence that TMC1 can contribute to a resting "leak" conductance that is in many ways distinct from the MET. Most striking is the lack of MTSET effects on TMC1 cysteine mutants thought to reside within the pore; in particular, MTSET does not alter the leak but reduces the transduction current in Tmc1 ko OHCs expressing TMC1-M412C. This is a remarkable finding and it is hard to understand how this could occur if these amino acids are part of a pore formed by TMC1. The experiments have been conducted carefully and the inclusion of long duration recording traces are particularly impactful and illustrate the consequences of the manipulations particularly clearly. There are however multiple issues with the work that need clarification before publication. The majority of the comments are related to presentation and interpretation, which make it difficult to appreciate the overall significance of the findings, given that the majority of the experiments focus on excitability of OHCs.

Essential revisions:

1) The writing and grammar in the manuscript needs a complete overhaul. There are many mistakes and unclear wordings which interfere with understanding what the authors are trying to convey.

2) Figure 1 panel B, what do the light gray and pink traces represent? (This is in most of the figures and should be stated at least in the first legend). I_m_ and I_BG_ should be defined directly, particularly in the legend. It is unclear the point of Figure 1G, but it should include an estimate of the reversal potential for each conductance. Do you know that the permeation of NMDG is the same or different between conductances?

3) The authors should provide better justification that either the endogenous or overexpressed leak is coming from the hair bundle. Might these channels be on the soma?

4) The mutagenesis data suggest TMC1 is part of the leak conductance, but the authors should not overstate the evidence to indicate that it is the channel. It might be a component and that can be suggested, but until the protein can be shown to actually be a channel, a more conservative approach to the writing would make sense. Basing conclusions on work that, in itself is not yet complete, weakens the argument. Given that the subunit composition and pore structure of the native channels will impact how the cysteine mutations manifest, these data should be used to highlight that the two conductances can be separated rather than used as evidence that it is the channel.

5) Descriptions of resting membrane current are only relevant when discussing what membrane potential the hair cells were held at. The recording configuration should also be described as whole-cell voltage clamp, not "patch-clamp" as patches of membrane were not isolated for these experiments.

6) It is not clear why the authors make the statement that "the inward I_BG_ was dramatically reduced in Tmc1-knockout OHCs", as both inward and outward components are reduced to nearly similar degrees in Tmc1-ko OHCs.

7) Statements about current amplitudes should include the data (average + SD/SEM and N values). The authors report current amplitudes significant to two decimal places, that is to the level of femtoamperes – this seems unnecessary, given the noise level of the recordings (5-10 pA, Figure 1B). Here and elsewhere-four significant digits makes no sense (unless the measurement is so precise that the errors justify it). Please report the error (SD or SEM, with n), keep the error to two significant digits, and adjust the significant digits in the mean accordingly.

8) Statistical analysis – the authors exclusively perform Student's T-tests, which it would be more appropriate to perform ANOVA followed by post-hoc test, to assess multiple comparisons (e.g. Figure 2D, Figure 6B, etc).

9) Why was the current amplitude so much smaller for the P3+ 1 OHCs than the acute P6 OHCs? It is notable that the resting current measured at P3 +1 is not different from the P6 TMC ko OHCs. Could the transfection be affecting the health of the cells?

10) "These data indicated that hair cells possess a background leak conductance, conferred specifically by TMC1." Given the complexity (and controversy) surrounding conductances in hair cells, I feel that this statement is too strong. It certainly suggests that TMC1 is necessary for the leak, but does not show it mediates it directly. In this regard, the lack of MTSET sensitivity could be regarded as refuting this hypothesis.

11) "Exogenously expressed TMC2 was significantly located in hair bundle of OHCs, as shown by HA tag (Figure 2—figure supplement 1)." I don't find this statement compelling based on the fluorescent traces provided. Images with better optical sectioning should be provided. The use of the word "significantly" should be backed by a quantitative assessment.

12) "Thus, neither mechanical nor pharmacological blockade of the MET channel affect I_Leak_ " This statement is also too strong. What the experiments show is that there is a holding current that persists when MET channels are closed or blocked with DHS. The experiments do not directly assess whether the leak current is affected by these manipulations.

13) Most striking is the MTSET effect – it does not alter the leak but reduces the transduction current in Tmc1 ko OHCs expressing TMC1-M412C. However, the high variability in I_Leak_ measurements decrease the power of the conclusions regarding the ability of different cysteine mutations to restore the leak current (particularly for M412C, T532C). What accounts for the high variance in this measurement? It is surprising that 5 out of the 6 cysteine mutants failed to restore the leak current. It is hard to reconcile that result with the idea that TMC1 contributes its pore both to the MET conductance and to the leak conductance. It is equally strange that M412C behaved identically to WT. Further discussion/clarification on these points is necessary.

14) Panel C in Figure 4—figure supplement 1 is not explained in the figure legend and there are no n values for these experiments.

15) To assess excitability, the authors move to IHCs, but there is insufficient evidence of TMC1 dependent leak current with similar characteristics in IHCs. The authors should indicate what age and location (along the cochlea) was used for these recordings.

16) The discussion of excitability changes was difficult to follow. In particular, the statements "the V_m_ was largely hyperpolarized" should be supported by data for clarity and the statement "Although the V_m_ in the non-bursting state was more hyperpolarized than in the bursting state in wild-type IHCs, it was positive to the V_m_ in Tmc1-knockout IHCs (Figure 7A,B)." needs to be revised.

17) The change in firing behavior TMC1 ko IHCs is difficult to understand. Presumably, these cells have a higher membrane resistance, so all other things being equal, they should experience greater depolarization for a given current injection, therefore increasing their excitability. As described, this is opposite to the phenotype observed. To control for resting state changes, these experiments should be repeated by measuring input output at the same membrane potential (neglecting the amount of current required to hold the cells at the same potential).

18) The authors make the statement "The MET current decreased from apex to base in Tmc1-knockout OHCs, which might result from that the hair bundle got more disrupted at base coil." If there could be differences in MET current amplitudes due to damage, this needs to be explored more thoroughly and accounted for, as it could complicate interpretation of the current amplitude gradients.

19) The variation in Ca sensitivity between MET and leak currents (Figure 8) is very perplexing and the authors do not provide a compelling answer to the observed variation.

20) The authors discuss three measured or derived currents: I-membrane, I-background, and I-leak. The writing and figures make it very difficult to figure out exactly what they are referring to, especially the difference between I-background and I-leak. I figured it out, but it took work. The distinction between these currents needs to be made much more clear.

[Editors' note: further revisions were requested prior to acceptance, as described below.]

Thank you for resubmitting your work entitled "TMC1 Is an Essential Component of a Leak Channel Underlying Tonotopy and Excitability of Auditory Hair Cells in Mice" for further consideration at *eLife*. Your revised article has been favorably evaluated by Richard Aldrich (Senior Editor), a Reviewing Editor, and two reviewers.

The manuscript has been improved but there are some remaining issues that need to be addressed before acceptance, as outlined below:

1) The general writing of the manuscript is still an obstacle to understanding the work. It needs major grammatical check as well as structural work to increase clarity and interpretation. For the revision, please send a version of the file with changes incorporated (without the Word edits highlighted). It is difficult to read the manuscript in this format.

2) The description of the contribution of TMC1 to the MET and leak currents needs further revision. For example, in the Discussion section: "Hence, we suggest a working model in which TMC1 integrates with the MET channel and a leak channel, and tunes the activity of hair cells." the authors should be more explicit about their thinking-it's not that TMC1 can be a leak channel or a MET channel, but it can incorporate into each. That makes interpretation of the results-especially the striking differences in behavior of the mutants-more understandable. This is an important point as some may have been thinking that the transduction channel (and the leak channel, by parallel) is just the TMCs, but it must be more complicated.

3) Subsection “The leak current modulates action potential firing in IHCs”. For the Tmc1 KO effects on excitability, how do we distinguish a reduction of MET resting Popen from the reduction in the leak current?

4) Subsection “TMC1 but not TMC2 mediates a background current in hair cells”. Change "the I_m_ was little (Figure 1B)…" to "the I_m_ was small (Figure 1B),…" (changing the word and adding a comma).

5) Subsection “TMC1 but not TMC2 mediates a background current in hair cells”. Change to "falls after P4".

6) Subsection “TMC1-mediated leak current is not carried by the resting open MET channel”. Change to something like: "Due to existing tension in the hair bundle, the open probability of MET channels at rest in hair cells is significant…"

7) Throughout. Unless it is used as a compound adjective, "amino-acid" or "amino-acids" should be changed to "amino acid" and "amino acids."

8) Subsection “Amino-acid substitutions in TMC1 alter the TMC1-mediated leak current”. The clause "neglecting to express any of the six cysteine-substituted TMC1 constructs" doesn't make sense.

9) Subsection “The leak current modulates action potential firing in IHCs”. The usual vernacular is to "depolarize the membrane," rather than "depolarize the membrane potential."

10) I'm not 100% sure, but I didn't think worm TMC-1 and TMC-2 are specifically homologous to (respectively) TMC1 and TMC2.

11) Discussion section. I don't understand this sentence: "However, it is still not yet explained that only a quarter proportion of TMC1 proteins are localized at the MET site, whereby counting around 2 MET complexes exist…"

12) The interpretation of the IV curves needs to be clarified. I would assume that the IV curves represent complex currents, not just leak and MET but other voltage gated channels as well. Would further assume the outward currents are not affected by NMDG but rather by the internal ions. So how these plots are interpreted is important and might be different depending on the voltage range that is investigated.

[Editors' note: further revisions were requested prior to acceptance, as described below.]

Thank you for resubmitting your work entitled "TMC1 Is an Essential Component of a Leak Channel Underlying Tonotopy and Excitability of Auditory Hair Cells in Mice" for further consideration by *eLife*. Your revised article has been evaluated by Richard Aldrich (Senior Editor) and a Reviewing Editor.

The manuscript has been improved but there are some minor revisions that are necessary for clarity, as outlined below:

Title:

TMC1 Is an Essential Component 1 of a Leak Channel Underlying Tonotopy and Excitability of Auditory Hair Cells in Mice

Change "underlying" to "that modulates".

Abstract:

Change to "The leak conductance is graded in frequency-dependent manner along the length of the cochlea…".

our results showed – change to "show".

Introduction:

"it is a pore-forming subunit" – define what "it" is.

Results:

It is noteworthy to point out that, I_BG_ was used as the title of vertical axis to show the dosage effect (Figure 5), but it represented the leak component when the concentration of blockers was high enough.

Revise – it is not clear which leak component you are referring to:

A significant leak conductance would be expected in order to depolarize the membrane and affect cell excitability,

Revise – "would be expected to depolarize…",

---

## [Author Response]

Summary:This is an interesting study which examined the contribution of TMC1 to the resting membrane "leak" conductance of hair cells in the mammalian cochlea. […] The majority of the comments are related to presentation and interpretation, which make it difficult to appreciate the overall significance of the findings, given that the majority of the experiments focus on excitability of OHCs.

Thank you for the constructive comments and suggestions that helped us to greatly improve the manuscript. We have conducted the requested experiments and integrated them in the revised manuscript.

Essential revisions:1) The writing and grammar in the manuscript needs a complete overhaul. There are many mistakes and unclear wordings which interfere with understanding what the authors are trying to convey.

We have sent the manuscript to a professional English editor for a review of the overall writing and grammar.

2) Figure 1 panel B, what do the light gray and pink traces represent? (This is in most of the figures and should be stated at least in the first legend). I_m_ and I_BG_ should be defined directly, particularly in the legend. It is unclear the point of Figure 1G, but it should include an estimate of the reversal potential for each conductance. Do you know that the permeation of NMDG is the same or different between conductances?

In Figure1B, the light gray and pink traces represent the low-pass filtered curves from original recordings that are noisier, which was done using Igor software and has been stated in Figure 1 legend, as well as in the Materials and methods section in the revised manuscript.

We updated the names of membrane currents to I_m_, I_Na_, I_NMDG_, and I_BG._ They are defined in the text and Figure 1 legend. “For more accurate quantification, the amplitude of the background current (I_BG_) was calculated by subtracting the I_m_ in 144 NMDG solution (I_NMDG_) from that in Na^+^ solution (I_Na_)”, which can be found in subsection” TMC1 but not TMC2 mediates a background current in hair cells “and Figure 1 legend.

Our intention was to show the voltage dependence of I_BG_ in the original Figure 1G (new Figure 1I). The I-V traces were recorded before and after 144 NMDG treatment extracellularly to make this measurement. Hence, only inward I_BG_ was calculated. In the revised version, an estimated reversal potential for I_BG_ from wild-type and TMC1-KO mice has been shown in new Figure 1H.

In terms of NMDG permeation, we made a comparison in the new Figure 1D. It appears that, *Tmc1*-KO has less NMDG current, but it is difficult to determine whether it is the technical leak from the whole-cell recordings, or the NMDG-permeable current.

3) The authors should provide better justification that either the endogenous or overexpressed leak is coming from the hair bundle. Might these channels be on the soma?

To answer this question, we designed an experiment by whole-cell recording OHCs after removing their hair bundles. We hypothesized that, if the leak current reduced in major proportion, the leak is coming from the hair bundle. If not, then the leak is coming from the soma. Our result showed that, the leak current decreased to 10 pA in hair-bundle removed OHCs, compared to the 48 pA leak current from intact OHCs. This piece of evidence indicated that, the leak conductance likely comes from the hair bundle. We did not successfully achieve similar recordings in cultured and overexpressed OHCs. We thought this figure helped to strengthen the conclusion of the manuscript. Hence, it was embedded in the revised manuscript as Figure 3—figure supplement 1.

4) The mutagenesis data suggest TMC1 is part of the leak conductance, but the authors should not overstate the evidence to indicate that it is the channel. It might be a component and that can be suggested, but until the protein can be shown to actually be a channel, a more conservative approach to the writing would make sense. Basing conclusions on work that, in itself is not yet complete, weakens the argument. Given that the subunit composition and pore structure of the native channels will impact how the cysteine mutations manifest, these data should be used to highlight that the two conductances can be separated rather than used as evidence that it is the channel.

We have removed the statement from the manuscript to indicate that TMC1 is the channel for leak conductance. In the revised version, we explain that, the channel is an essential component for the leak channels. Accordingly, we focused on our discussion regarding that, the properties and function of the leak channel are distinct to that of the MET channel.

5) Descriptions of resting membrane current are only relevant when discussing what membrane potential the hair cells were held at. The recording configuration should also be described as whole-cell voltage clamp, not "patch-clamp" as patches of membrane were not isolated for these experiments.

We have changed the wording “whole-cell patch-clamp” to “whole-cell voltage-clamp” or “whole-cell current-clamp” based on the configuration used in each experiment.

6) It is not clear why the authors make the statement that "the inward I_BG_ was dramatically reduced in Tmc1-knockout OHCs", as both inward and outward components are reduced to nearly similar degrees in Tmc1-ko OHCs.

In the second part of Figure 1, we intended to measure the voltage dependence of I_BG_ based on I-V protocol. The original Figure 1E (new Figure 1F) only showed the traces before NMDG treatment. Currently, the I-V traces before and after NMDG treatment were shown in new Figure 1F. The I_BG_ was measured from membrane currents before and after NMDG treatment extracellularly. Hence, only inward current (new Figure 1I / original Figure 1G) was measured from the I-V recordings in new Figure 1F (original Figure 1E). We also changed the sentence to “After subtraction, it was clear that the I_BG_ altered with holding potentials and was dramatically reduced in *Tmc1*-knockout OHCs (Figure 1I)”. Now in subsection “TMC1 but not TMC2 mediates a background current in hair cells”.

7) Statements about current amplitudes should include the data (average + SD/SEM and N values). The authors report current amplitudes significant to two decimal places, that is to the level of femtoamperes – this seems unnecessary, given the noise level of the recordings (5-10 pA, Figure 1B). Here and elsewhere-four significant digits makes no sense (unless the measurement is so precise that the errors justify it). Please report the error (SD or SEM, with n), keep the error to two significant digits, and adjust the significant digits in the mean accordingly.

In the revised version, we described mean+/-SD/SEM, n values and applied statistical analysis in every figure legend. For example, in Figure 1 it says, “Data are presented as mean ± SEM. N values are shown in each panel. *p <0.05, **p <0.01, ***p <0.001, Student’s t-test”. We have changed all the current value to integers and made according adjustment to the values from other measurements (e.g. V_m_ in Figure 7).

8) Statistical analysis – the authors exclusively perform Student's T-tests, which it would be more appropriate to perform ANOVA followed by post-hoc test, to assess multiple comparisons (e.g. Figure 2D, Figure 6B, etc.).

In the revised manuscript, we applied statistical analysis based on the characteristics of the data set. For example, for one-to-one comparison we used student’s T-test, and for multiple comparisons we used ANOVA.

9) Why was the current amplitude so much smaller for the P3+ 1 OHCs than the acute P6 OHCs? It is notable that the resting current measured at P3 +1 is not different from the P6 TMC ko OHCs. Could the transfection be affecting the health of the cells?

We added experiments demonstrating that, P6 OHCs was cultured for 1 day in vitro (P6+1DIV) and whole-cell voltage-clamp was recorded. The leak current amplitude was approximately 64 pA (Author response image 1, left bar), relevant to that (71 pA) recorded from P6 OHCs in acutely dissociated cochlea tissue (Author response image 1, right bar, adapted from Figure 1E). Similarly, Figure 2 showed that leak current was 18 pA in P3+1DIV OHCs (Figure 2D) and the leak current was 19 pA in dissociated P3 OHCs (Figure 2F). From these data, 1-DIV culture did not obviously affect the leak function of the OHCs.

10) "These data indicated that hair cells possess a background leak conductance, conferred specifically by TMC1." Given the complexity (and controversy) surrounding conductances in hair cells, I feel that this statement is too strong. It certainly suggests that TMC1 is necessary for the leak, but does not show it mediates it directly. In this regard, the lack of MTSET sensitivity could be regarded as refuting this hypothesis.

We have changed the statement to, “These data indicated that, hair cells possess a background leak conductance that functionally couples with TMC1” in subsection “TMC1 but not TMC2 mediates a background current in hair cells”. We also made a statement explaining that, TMC1 is essential for the leak current, but the leak channel possesses different properties from the MET channel in terms of the role of TMC1.

11) "Exogenously expressed TMC2 was significantly located in hair bundle of OHCs, as shown by HA tag (Figure 2—figure supplement 1)." I don't find this statement compelling based on the fluorescent traces provided. Images with better optical sectioning should be provided. The use of the word "significantly" should be backed by a quantitative assessment.

We have again performed the immunostaining experiment on TMC2-HA transfected cochleae. This time, imaging with better optical effect was collected, which showed condensed HA staining on hair bundle layer (new Figure 2—figure supplement 1). We also changed the word, “significantly” to “visibly”. Subsection “TMC1 but not TMC2 mediates a background current in hair cells”.

12) "Thus, neither mechanical nor pharmacological blockade of the MET channel affect I_Leak_ " This statement is also too strong. What the experiments show is that there is a holding current that persists when MET channels are closed or blocked with DHS. The experiments do not directly assess whether the leak current is affected by these manipulations.

We changed the sentence to, “the I_Leak_ persisted in either mechanical closure or pharmacological blockade of the MET channel”. Subsection “TMC1-mediated leak current is not carried by the resting open MET channel”.

13) Most striking is the MTSET effect – it does not alter the leak but reduces the transduction current in Tmc1 ko OHCs expressing TMC1-M412C. However, the high variability in I_Leak_ measurements decreases the power of the conclusions regarding the ability of different cysteine mutations to restore the leak current (particularly for M412C, T532C). What accounts for the high variance in this measurement? It is surprising that 5 out of the 6 cysteine mutants failed to restore the leak current. It is hard to reconcile that result with the idea that TMC1 contributes its pore both to the MET conductance and to the leak conductance. It is equally strange that M412C behaved identically to WT. Further discussion/clarification on these points is necessary.

In this study, we described that, the leak current that was around 60 pA, less than that from most of general channels. Especially, under treatment or manipulation of hair cells (e.g. tissue culture status and TMC1 overexpression), the current amplitude varies depending on quantity of TMC1 proteins. Our solution was to make more recordings in order to gain better quantification from mean and distribution analysis. Based on amino-acid substitution experiment and other results, the leak channel demonstrated distinct properties from the MET channel, including the M412C that regulated the MET channel conductance but not the leak channel conductance. We agreed that, M412 is an interesting site and that M412C mutation did not affect the leak current, even with MTSET treatment. While previous evidence has shown that MTSET treatment of M412C reduces the MET current (Pan et al., 2018), and M412K causes deafness without affecting the MET current in Beethoven mice (Marcotti et al., 2006). In terms of the leak conductance, M412C behaved similar to wild-type, which is not atypical since other evidence has shown the distinct dosage sensitivities to factors (e.g. channel blockers and calcium, between the leak channel and the MET channel). In the revised manuscript, we stated that, TMC1 is essential for the leak channel, but it not yet ready to claim it as the pore for the leak channel. In the discussion, the difference between leak and MET channels are stated. Discussion section.

14) Panel C in Figure 4—figure supplement 1 is not explained in the figure legend and there are no n values for these experiments.

In the revised manuscript, the Figure 4—figure supplement 1. Panel C has been described in the figure legend and the n values were added

15) To assess excitability, the authors move to IHCs, but there is insufficient evidence of TMC1 dependent leak current with similar characteristics in IHCs. The authors should indicate what age and location (along the cochlea) was used for these recordings.

For the IHC recordings, the mouse age was P6 and the location recorded was 0.4 to the very apex as length of the whole cochlear coil was 1.0. At the same age and recording location of cochlea, the leak current amplitude was obviously different between IHCs (24 pA, n=11, previous Figure 7D) and OHCs (51 pA, n=9, new Figure 3E). We also added more IHC recordings to raise the n value from 11 to 20, with the leak current being 25 pA (new Figure 7D); similar to the previous value. We hypothesized that, it may have resulted from the less number of TMC1 proteins in IHCs compared to that in OHCs (Beurg et al., 2018), which is discussed. Subsection “The leak current modulates action potential firing in IHCs”.

16) The discussion of excitability changes was difficult to follow. In particular, the statements "the V_m_ was largely hyperpolarized" should be supported by data for clarity and the statement "Although the V_m_ in the non-bursting state was more hyperpolarized than in the bursting state in wild-type IHCs, it was positive to the V_m_ in Tmc1-knockout IHCs (Figure 7A,B)." needs to be revised.

The sentence "the V_m_ was largely hyperpolarized" was changed to “the resting V_m_ was more hyperpolarized” in subsection “The leak current modulates action potential firing in IHCs”. It was supported by the quantification in Figure 7B. The statement "Although the V_m_ in the non-bursting state was more hyperpolarized than in the bursting state in wild-type IHCs, it was positive to the V_m_ in Tmc1-knockout IHCs (Figure 7A,B)." was revised to “Although the V_m_ baseline in the non-bursting state was more hyperpolarized than that in the bursting state in wild-type IHCs, it was still more depolarized than the V_m_ baseline in *Tmc1*-knockout IHCs (Figure 7A,B)” subsection “The leak current modulates action potential firing in IHCs”.

17) The change in firing behavior TMC1 ko IHCs is difficult to understand. Presumably, these cells have a higher membrane resistance, so all other things being equal, they should experience greater depolarization for a given current injection, therefore increasing their excitability. As described, this is opposite to the phenotype observed. To control for resting state changes, these experiments should be repeated by measuring input output at the same membrane potential (neglecting the amount of current required to hold the cells at the same potential).

We added experiments to answer this question. By adjusting holding current, the membrane potential was brought to -60 mV in wild-type and *Tmc1*-knockout IHCs. When injecting additional 50 pA current, the firing rate of action potentials in *Tmc1*-knockout IHCs was higher than that in wild-type IHCs, as shown in Author response image 2. Hence, it is not opposite to the phenotype we described in Figure 7. Due to lack of the leak conductance that induces a higher membrane resistance in Tmc1-knockout IHCs than that in wild-type IHCs, the V_m_ is more hyperpolarized and action potential fires less in resting condition. Hence, when artificially bringing the holding V_m_ to an identical level followed by injecting the same amount of current, the action potential fires more in *Tmc1*-knockout IHCs.

**Author response image 2. respfig2:** 

18) The authors make the statement "The MET current decreased from apex to base in Tmc1-knockout OHCs, which might result from that the hair bundle got more disrupted at base coil." If there could be differences in MET current amplitudes due to damage, this needs to be explored more thoroughly and accounted for, as it could complicate interpretation of the current amplitude gradients.

The numbers of expressed TMC1 proteins could determine the amplitude of the MET current. In the revised manuscript, we stated that, “the leak and MET current decreased from apex to base in *Tmc1*-knockout OHCs, which correlated with the graded expression level of TMC1 along the cochlear coil” in subsection “The leak current follows the tonotopic gradient of the MET response in OHCs”. Moreover, we discussed the rationality in the Discussion section.

19) The variation in Ca sensitivity between MET and leak currents (Figure 8) is very perplexing and the authors do not provide a compelling answer to the observed variation.

As shown in Figure 6C,D, the P_Na_/P_Cs_ was 0.7 and the P_Ca_/P_Cs_ was 0, indicating Ca^2+^ barely entered the leak channel. This dataset could not be used to calculate Ca^2+^ permeability of the MET channel because we added DHS in the external solution. Thus, we included an experiment to measure the Ca^2+^ permeability of MET channel. The P_Na_/P_Cs_ is 1.3, while the P_Ca_/P_Cs_ is 6.4, which is consistent with the report by Kim and Fettiplace, (2013). Figure 6F showed a typical dosage-dependent inhibition of Ca^2+^ on the leak conductance but not on MET channel. Thus, in Figure 8D,E, 35 mM Ca^2+^ transiently abolished the leak current and tonotopic gradient of the MET current in wild-type OHCs, similar to Tmc1-deficiency induced effect. In the revised manuscript, we compared the difference between the MET channel and the leak channel in terms of the Ca^2+^ sensitivity in OHCs. We hypothesized that, Ca^2+^ may inhibit the leak channel, although it is permeable to the MET channel.

20) The authors discuss three measured or derived currents: I-membrane, I-background, and I-leak. The writing and figures make it very difficult to figure out exactly what they are referring to, especially the difference between I-background and I-leak. I figured it out, but it took work. The distinction between these currents needs to be made much more clear.

We introduced a series of values for further measurement in the revised manuscript. They are I_m_ (recorded membrane current), I_Na_ (I_m_ under 144 Na recording solution), I_NMDG_ (I_m_ under 144 NMDG treatment), I_BG_ (background current by subtracting I_NMDG_ from I_Na_), I_DHS_ (I_m_ under 100 μM DHS treatment), and I_Leak_ (resting-MET independent leak current). In Figure 1 legend, it says, “I_BG_ (background current) was calculated by subtraction of I_m_ in 144 Na (I_Na_) and I_m_ in 144 NMDG (I_NMDG_) to exclude technical leak”. In text, it says, “For more accurate quantification, the amplitude of the background current (I_BG_) was calculated by subtracting the I_m_ in 144 NMDG solution (I_NMDG_) from that in Na^+^ solution (I_Na_)”, subsection “TMC1 but not TMC2 mediates a background current in hair cells”. In Figure 3 legend, it says, “The baseline current was similar when the MET channels were closed by either FJ (#1) or DHS (I_DHS_, #2), as highlighted with a red dashed line. As shown in #3, the DHS-sensitive resting MET current (I_Resting-MET_) was calculated by subtraction of I_Na_ and I_DHS_. The baseline current was further closed by NMDG, defined as I_Leak_ by subtraction of I_DHS_ and I_NMDG_.”.

[Editors' note: further revisions were requested prior to acceptance, as described below.]

The manuscript has been improved but there are some remaining issues that need to be addressed before acceptance, as outlined below:1) The general writing of the manuscript is still an obstacle to understanding the work. It needs major grammatical check as well as structural work to increase clarity and interpretation. For the revision, please send a version of the file with changes incorporated (without the Word edits highlighted). It is difficult to read the manuscript in this format.

We have gone through a major grammatical check and made some structural work to improve the clarity and interpretation.

2) The description of the contribution of TMC1 to the MET and leak currents needs further revision. For example, in the Discussion section: "Hence, we suggest a working model in which TMC1 integrates with the MET channel and a leak channel, and tunes the activity of hair cells." the authors should be more explicit about their thinking-it's not that TMC1 can be a leak channel or a MET channel, but it can incorporate into each. That makes interpretation of the results-especially the striking differences in behavior of the mutants-more understandable. This is an important point as some may have been thinking that the transduction channel (and the leak channel, by parallel) is just the TMCs, but it must be more complicated.

We have updated the last paragraph to clarity our opinion in the Discussion section. For example, “Hence, we suggest a working model in which TMC1 functionally incorporates into a leak channel and the MET channel, rather than being the pore of the two channels, and tunes the activity of hair cells.”

3) Subsection “The leak current modulates action potential firing in IHCs”. For the Tmc1 KO effects on excitability, how do we distinguish a reduction of MET resting Popen from the reduction in the leak current?

In Figure 7, the IHCs were recorded in the external solution with 100 μM dihydrostreptomycin that was used to block the resting and/or evoked MET currents. We changed the sentence in subsection “The leak current modulates action potential firing in IHCs” to “Therefore, we measured the membrane potential (V_m_) bathed in IHCs in external solution with 100 μM DHS by whole-cell current-clamp recording (Figure 7A).” and the sentences to “(A) Representative current-clamp recording in IHCs bathed in external solution with 100 μM DHS from wild-type (black) and Tmc1-knockout (red) mice.” on line 484 and “In this figure, the external solution contained 1.3 mM Ca^2+^ and 100 μM DHS in figure legend accordingly.

4) Subsection “TMC1 but not TMC2 mediates a background current in hair cells”. Change "the I_m_ was little (Figure 1B)…" to "the I_m_ was small (Figure 1B),…" (changing the word and adding a comma).

It has been changed to “the I_m_ was small (Figure 1B),”.

5) Subsection “TMC1 but not TMC2 mediates a background current in hair cells”. Change to "falls after P4".

It has been changed to “falls after P4”.

6) Subsection “TMC1-mediated leak current is not carried by the resting open MET channel”. Change to something like: "Due to existing tension in the hair bundle, the open probability of MET channels at rest in hair cells is significant…"

It has been changed.

7) Throughout. Unless it is used as a compound adjective, "amino-acid" or "amino-acids" should be changed to "amino acid" and "amino acids."

In the text, most of expression “amino-acid” has been changed to “amino acid”, excepting in several place “amino-acid” has been used as a compound adjective.

8) Subsection “Amino-acid substitutions in TMC1 alter the TMC1-mediated leak current”. The clause "neglecting to express any of the six cysteine-substituted TMC1 constructs" doesn't make sense.

We change the sentence to “did not, however, change the current baseline in OHCs when expressing any of the six cysteine-substituted TMC1 constructs”.

9) Subsection “The leak current modulates action potential firing in IHCs”. The usual vernacular is to "depolarize the membrane," rather than "depolarize the membrane potential."

It has been changed to “depolarize the membrane”.

10) I'm not 100% sure, but I didn't think worm TMC-1 and TMC-2 are specifically homologous to (respectively) TMC1 and TMC2.

We have deleted the sentence “It also indicates different TMC functions crossing species; as it has been proposed that both TMC-1 and TMC-2 confer the leak conductance in worms” in the Discussion section.

11) Discussion section. I don't understand this sentence: "However, it is still not yet explained that only a quarter proportion of TMC1 proteins are localized at the MET site, whereby counting around 2 MET complexes exist…"

It has been changed to “However, not all TMC1 proteins are localized at the MET site, where only up to 2 MET channel complexes exist” in the Discussion section.

12) The interpretation of the IV curves needs to be clarified. I would assume that the IV curves represent complex currents, not just leak and MET but other voltage gated channels as well. Would further assume the outward currents are not affected by NMDG but rather by the internal ions. So how these plots are interpreted is important and might be different depending on the voltage range that is investigated.

Yes, I_m_ represents complex currents that include leak current, resting MET current, and other channels. Because NMDG was applied only extracellularly, the inward currents but not outward currents were affected by NMDG. Then we only calculated the inward background current (I_m_-I_NMDG_) from recordings similar to Figure 1F and I_BG_ was used in Figure 1 and Figure 2. However, as you mentioned that the I_background_ was still complex. According to the investigation requirement, we then used I_Leak_ to describe the leak current that may mostly represent the contribution by TMC1 (Figure 3, Figure 4, Figure 5, Figure 6, Figure 7 and Figure 8). The aim of I-V experiments was to show the voltage dependence of these inward background currents. We have updated sentence in subsection “TMC1 but not TMC2 mediates a background current in hair cells” to “Furthermore, the voltage dependence of I_m_ and I_NMDG_ was analyzed by applying a series of voltage-pulse stimuli to OHCs (Figure 1F-I). The I_m_-V curves obtained from these measurements verified a reduced I_m_ (Figure 1G) and a more negative reversal potential (Figure 1H) in Tmc1-knockout OHCs. After subtraction (only inward I_BG_ was calculated because NMDG was applied extracellularly), it was clear that the I_BG_ altered almost linearly with holding potentials and was dramatically reduced in Tmc1-knockout OHCs (Figure 1I).”

[Editors' note: further revisions were requested prior to acceptance, as described below.]

The manuscript has been improved but there are some minor revisions that are necessary for clarity, as outlined below:Title:TMC1 Is an Essential Component 1 of a Leak Channel Underlying Tonotopy and Excitability of Auditory Hair Cells in MiceChange "underlying" to "that modulates".

In the revised manuscript, “underlying” has been changed to “that modulates” in the title.

Abstract:Change to "The leak conductance is graded in frequency-dependent manner along the length of the cochlea…".our results showed – change to "show".

In the revised version, “The leak conductance is graded in frequency-dependent manner…” has been changed to “The leak conductance is graded in frequency-dependent manner along the length of the cochlea…” on page 2, line 30 and “showed” has been changed to “show” in the Abstract.

Introduction:"it is a pore-forming subunit" – define what "it" is.

In the revised version, “it” has been changed to “TMC1” in the Introduction.

Results:It is noteworthy to point out that, I_BG_ was used as the title of vertical axis to show the dosage effect (Figure 5), but it represented the leak component when the concentration of blockers was high enough.Revise – it is not clear which leak component you are referring to:

In the revised version, this sentence has been updated to “It is noteworthy to point out that the I_BG_ included the I_Resting-MET_ and the I_Leak_ when the concentration of the blockers was low, but the I_BG_ was mainly composed of the I_Leak_ when the concentration of blockers was high enough (Figure 5).” in subsection “Pharmacological blockade of the TMC1-mediated leak conductance”.

A significant leak conductance would be expected in order to depolarize the membrane and affect cell excitability,Revise – "would be expected to depolarize…",

In the revised version, “would be expected to in order to depolarize…” has been updated to “would be expected to depolarize…” in subsection “The leak current modulates action potential firing in IHCs”.